# Specific lexico-semantic predictions are associated with unique spatial and temporal patterns of neural activity

**Lin Wang[1,2,3]\*, Gina Kuperberg[1,2,3], Ole Jensen[4]**

[1]Department of Psychiatry, Harvard Medical School, Boston, United States;
[2]Athinoula A. Martinos Center for Biomedical Imaging, Massachusetts General Hospital, Charlestown, United States; [3]Department of Psychology, Tufts University, Medford, United States; [4]School of Psychology, Centre for Human Brain Health, University of Birmingham, Birmingham, United Kingdom

**Abstract** We used Magnetoencephalography (MEG) in combination with Representational Similarity Analysis to probe neural activity associated with distinct, item-specific lexico-semantic predictions during language comprehension. MEG activity was measured as participants read highly constraining sentences in which the final words could be predicted. Before the onset of the predicted words, both the spatial and temporal patterns of brain activity were more similar when the same words were predicted than when different words were predicted. The temporal patterns localized to the left inferior and medial temporal lobe. These findings provide evidence that unique spatial and temporal patterns of neural activity are associated with item-specific lexico-semantic predictions. We suggest that the unique spatial patterns reflected the prediction of spatially distributed semantic features associated with the predicted word, and that the left inferior/medial temporal lobe played a role in temporally 'binding' these features, giving rise to unique lexico-semantic predictions.

DOI: https://doi.org/10.7554/eLife.39061.001

**\*For correspondence:**
wanglinsisi@gmail.com

**Competing interests:** The authors declare that no competing interests exist.

## Introduction

After reading or hearing the sentence context, 'In the crib there is a sleeping . . .', we are easily able to predict the next word, 'baby'. In other words, we are able to access a unique lexico-semantic representation of <baby> that is different from the lexico-semantic representation of any other word (e.g. <rose>), ahead of this information becoming available from the bottom-up input. In the present study, we used Magnetoencephalography (MEG), in combination with Representational Similarity Analysis (RSA), to show that the prediction of specific words is associated with distinct spatial and temporal patterns of neural activity before the predicted word is actually presented.

Prediction is hypothesized to be a core computational principle of brain function (*Clark, 2013*; *Mumford, 1992*). During language processing, probabilistic prediction at multiple levels of representation allows us to rapidly understand what we read or hear by giving processing a head start (see *Kuperberg and Jaeger, 2016a*, for a review). The strength of prediction, and the precise level of representation at which it occurs, is likely to depend on many factors (see *Kuperberg and Jaeger, 2016a*, section 3.4). However, there is now clear neural evidence that, at least in highly constraining sentence contexts, we are able to predict the semantic features of upcoming words.

This evidence comes from several sources. First, a large body of studies show that the N400 — an event related potential (ERP) that reflects semantic processing — is reduced in response to words whose semantic features match semantic predictions generated by highly predictable (versus less predictable) contexts. For example, the N400 elicited by 'baby' is smaller in the constraining context

'In the crib, there is a sleeping . . .' than in the less constraining context, 'Under the tree, there is a sleeping . . .' (*Federmeier and Kutas, 1999*; *Kutas and Federmeier, 2011*; *Kuperberg, 2016b*).

Second, several studies have reported differential modulation of brain activity following highly predictable versus less predictable sentence contexts, prior to the onset of predicted words. These include larger negative-going ERP effects (*Freunberger and Roehm, 2017*; *Grisoni et al., 2017*; *León-Cabrera et al., 2017*; *Maess et al., 2016*), increases in theta power (*Dikker and Pylkkänen, 2013*; *Piai et al., 2016*), and the suppression of alpha/beta power (*Piai et al., 2014*; *Piai et al., 2015*; *Rommers et al., 2017*; *Wang et al., 2018*). These anticipatory effects have been neuroanatomically localized to both neocortical (e.g. left frontal and temporal regions; *Dikker and Pylkkänen, 2013*; *Piai et al., 2015*; *Wang et al., 2018*) and subcortical (e.g. hippocampus and cerebellum; *Bonhage et al., 2015*; *Lesage et al., 2017*; *Piai et al., 2016*; *Wang et al., 2018*) regions. They have been attributed either to the process of generating predictions and/or access to the lexico-semantic representations that correspond to predicted words themselves. Importantly, however, previous studies have averaged across items that predict *different* upcoming words. It therefore remains unclear whether the brain produces unique patterns of neural activity that correspond to the prediction of *item-specific* lexico-semantic representations. For example, does the particular pattern of neural activity that is produced following the context, 'In the crib there is a sleeping . . .' differ from the pattern of neural activity produced following the context, 'On Valentine's day, he sent his girlfriend a bouquet of red . . .'?

Multivariate Pattern Analysis (MVPA) provides one way of addressing this question (*Kriegeskorte et al., 2008*; *Staudigl et al., 2015*; *Stokes et al., 2015a*). Correlational approaches were first applied to fMRI data to identify spatial patterns of brain activity representing objects categories in the ventral stream (*Haxby et al., 2001*). They later evolved into Representational Similarity Analysis (RSA). The basic assumption of RSA is that similarities in patterns of brain activity reflect similarities between representationally similar items. Spatial RSA has been used to identify unique patterns of spatial activity during perception, cognition and action (*Kriegeskorte et al., 2007*; *Kriegeskorte and Kievit, 2013*). More recently, it has been applied to MEG and EEG data whose excellent temporal resolution can tell us exactly *when* such spatially-specific patterns of neural activity are activated in relation to the appearance of bottom-up input (*Stokes et al., 2015a*). Moreover, the precise temporal resolution of MEG/EEG also allows for the use of an analogous RSA approach that probes *temporal* rather than *spatial* patterns of neural similarity (*Staudigl et al., 2015*; *Michelmann et al., 2016*). Both spatial and temporal RSA approaches have been successfully used in combination with MEG and EEG to decode representationally specific visual information during the perception of bottom-up input (*Cichy et al., 2014*), as well as during its maintenance in working memory in the absence of bottom-up input (*Wolff et al., 2017*).

In the present study, we used MEG, together with both spatial and temporal RSA, to ask whether, under experimental conditions that are known to encourage specific lexico-semantic prediction, distinct words are associated with distinct spatial and temporal patterns of neural activity, prior to the appearance of the predicted input. Participants read 240 sentences, all with highly constraining contexts that predicted a specific word (*Figure 1A*). The sentences were visually presented at a slow rate of 1000 ms per word. This ensured the generation of specific lexico-semantic predictions and guaranteed sufficient time to detect any representationally specific neural activity before the onset of the predicted word. We constructed these sentences in pairs (120 pairs) such that each member of a pair predicted the same word, even though their contexts differed (e.g. 'In the crib, there is a sleeping . . .' and 'In the hospital, there is a newborn . . .'). During the experiment, sentences were presented in a pseudorandom order, with at least 30 other sentences (on average 88 sentences) in between each member of a given pair.

There is some evidence that the various semantic features and properties associated with words and concepts are represented in the brain across spatially distributed multimodal networks (*Damasio, 1989*; *Price, 2000*; *Martin and Chao, 2001*), which can be detected using spatial RSA (e.g. *Devereux et al., 2013*). For example, the particular set of semantic features and properties associated with the concept, <baby> (e.g. <human>, <small>, <cries>), might be represented by a particular spatially distributed pattern of neural activity, whereas the semantic features and properties associated with the concept, <rose> (e.g. <plant>, <scalloped petals>, <fragrant smell>) might be represented by a different spatially distributed pattern of neural activity. If, following a constraining context (e.g. In the crib, there is a sleeping . . .'), the prediction of a unique lexico-semantic item

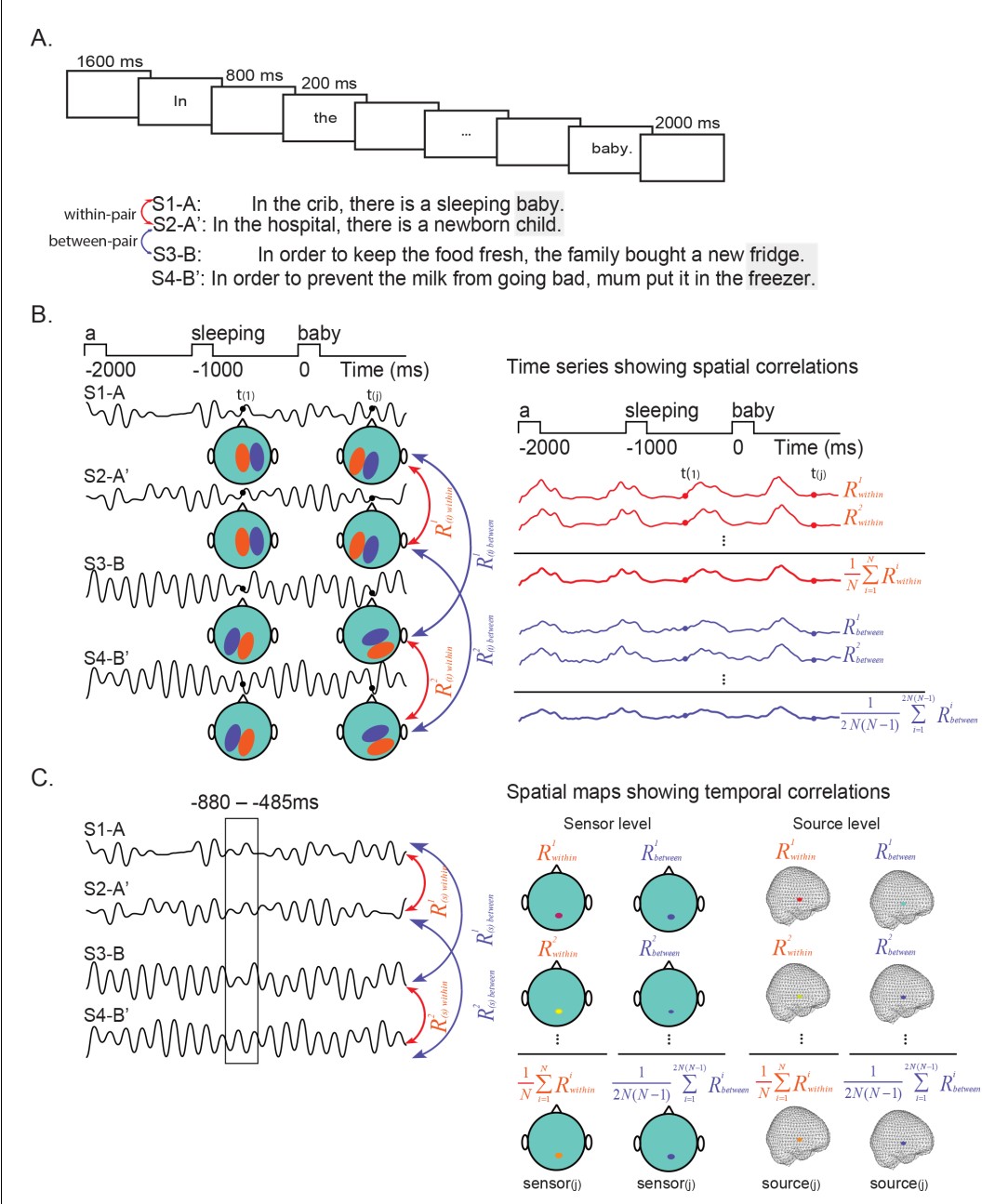

**Figure 1.** The experimental procedure and approach for Representational Similarity Analyses. (**A**) Trials began with a blank screen (1600 ms). Sentences were presented in Chinese (translated here into English), word-by-word (200 ms per word; 800 ms blank interval between words). Sentences were followed either by 'NEXT' (2000 ms) or by a probe question (1/6th of trials, randomly). We constructed sentences in pairs such that the same word could be predicted from the context (e.g. S1-A and S2-A'; S3-B and S4-B') (although during presentation, members of each pair were presented separately, with at least 30 other sentences in between). One member of each pair ended with the predicted word (e.g. S1–A, S3–B) and the other member ended with a plausible but unpredicted word (e.g. S2–A', S4–B'). Before the onset of the predicted word, we compared brain activity associated with the prediction of the same word (*within-pairs*) and a different word (*between-pairs*). (**B**) Spatial representational similarity analysis. Left: The pattern of MEG data over sensors was correlated between each sentence pair (e.g. S1–A and S2–A') at each time sample $t_{(j)}$. Right: The average spatial correlation values of pairs ($R^1_{within}$, $R^2_{within}$, ...) in which the same word was predicted formed the *within-pair* spatial correlation time series ($\frac{1}{N}\sum_{i=1}^{N} R^i_{within}$, shown in red). The average spatial correlation values of pairs ($R^1_{between}$, $R^2_{between}$, ...) in which different words were predicted formed the *between-pair* spatial correlation time series ($\frac{1}{2N(N-1)}\sum_{i=1}^{2N(N-1)} R^i_{between}$, shown in blue). (**C**) Temporal representational similarity analysis. Left: The temporal pattern of MEG activity was correlated between sentence pairs, at each sensor (sensor space) or at each grid point (source space). Right: The average temporal correlation

*Figure 1 continued on next page*

*Figure 1 continued*

values of pairs ($R^1_{within}$, $R^2_{within}$, . . .) in which the same word was predicted formed the *within-pair* temporal correlation topographic/source maps. The average temporal correlation values of pairs ($R^1_{between}$, $R^2_{between}$, . . .) in which different words were predicted formed the *between-pair* temporal correlation topographic/source maps.

DOI: https://doi.org/10.7554/eLife.39061.002

The following source data is available for figure 1:

**Source data 1.** Chinese stimuli together with their English translations, as well as some measures for the context up until the sentence final word (SFW), the word before the sentence final word (SFW-1) and the sentence final words (SFW).

DOI: https://doi.org/10.7554/eLife.39061.003

(<baby>) is represented by a unique spatial pattern of brain activity, then this spatial pattern should be more similar following another context that predicts the same word, that is *within-pair* (e.g. 'In the hospital, there is a newborn . . .') than following another context that predicts a different word, that is *between-pair* (e.g. On Valentine's day, he sent his girlfriend a bouquet of red . . .'). This should be just as true if we average across all *within-pair* sentences and compare them with all *between-pair* sentences. Importantly, this effect should be evident prior to the onset of the predicted word.

To test this hypothesis, we correlated the spatial pattern of MEG data across all sensors, between all possible pairs of sentences, at all time points over the last three words of the sentences. We were particularly interested in activity following the first word at which specific lexico-semantic predictions of upcoming words could be generated: the word before the SFW. We asked whether, at this point, the resulting spatial similarity values were larger following sentence contexts that constrained for the same word (*within-pairs*) than those that constrained for a different word (*between-pairs*), see *Figure 1B*.

A classic hypothesis of how spatially distributed semantic information becomes bound together to represent specific concepts in the brain is through a process of 'temporal synchrony' (*Damasio, 1989*). If, following a highly constraining context, the prediction of a unique lexico-semantic item is instantiated through a unique *temporal* pattern of brain activity, then the temporal patterns of neural activity should be more similar following pairs of sentence contexts that constrain for the same word (*within-pairs*) than following pairs that constrain for a different word (*between-pairs*). To test this hypothesis, we correlated the temporal pattern of MEG data evoked within the prediction period (before the onset of predicted word) between all possible pairs of sentences at each MEG sensor, and we asked whether there were any sensors in which the resulting temporal similarity values were larger for *within-pair* sentences than *between-pair* sentences.

*Damasio (1989)* also hypothesized that temporal binding occurred within so-called 'convergence zones' of the brain. Although it is still a matter of debate whether multiple convergence zones exist, parts of the temporal lobe, including anterior, ventral and medial regions, have been identified as 'semantic hubs' that bring spatially distributed semantic information together to form single concepts (*Patterson et al., 2007*; *Ralph et al., 2017*). If these regions play a functional role in instantiating the prediction of specific lexico-semantic items through temporal binding, then unique temporal patterns of prediction (*Figure 1C*) should localize to these regions. To test this hypothesis, we used source localization techniques to determine the neuroanatomical source of any increased temporal similarity following sentence contexts that predicted the same word (*within-pairs*) versus a different word (*between-pairs*).

## Results

Twenty-six participants read 240 sentences, presented at a rate of one word per second, while MEG data were acquired. The sentences were constructed in pairs (120 pairs) that strongly predicted the same sentence-final word (SFW), although, during presentation, members of the same pair were separated by at least 30 (on average 88) other sentences. As an example (*Figure 1A*), sentences S1 (e.g. 'In the crib there is a sleeping. . .') and S2 ('In the hospital, there is a newborn. . .') both predicted the word 'baby'. To avoid repetition of the predicted word across sentence pairs, one member of each pair ended with the predicted word (e.g. in S1, 'baby') while the other member ended with an unpredicted but plausible word (e.g. in S2, 'child'). Participants were asked to read each sentence carefully and to answer yes/no comprehension questions following 1/6th of the sentences.

Comprehension accuracy was high (98% ± 2.0%). We compared both spatial and temporal similarity patterns of sentence pairs that predicted the same SFWs (*within-pairs*) to those that predicted different SFWs (*between-pairs*), before the SFW actually appeared. All trials were included in the analysis.

## Spatial RSA: The spatial pattern of neural activity was more similar in sentence pairs that predicted the same versus different words, and this effect began before the onset of the predicted word

In each participant, we quantified the degree of spatial similarity of MEG activity (30 Hz low-pass filter) produced by pairs of sentences that predicted either the same SFW (i.e. *within-pairs*, for example S1-A vs. S2-A') or a different SFW (i.e. *between-pairs*, for example S1-A vs. S3-B) by correlating the spatial pattern of signal across sensors at each sampling point from −2000 ms until 1000 ms, relative to the onset of the SFW. We then averaged the resulting time series of spatial correlations (R-values), first within each participant and then across participants (*Figure 2B*). Both the *within-* and *between-pair* group-averaged time series of spatial correlation values showed a sharp increase at ~100 ms after the onset of the penultimate word (SFW-1; at −1000 ms) that lasted ~400 ms before sharply decreasing again (*Figure 2A*). The same pattern was observed around the onset of the previous word (SFW-2) and around the onset of the SFW itself. We attribute this general increase in spatial similarity to the visual onset and offset of each word. This general increase in spatial correlation was largest between −880 − −485 ms (R > 0.04) before the onset of the SFW, and between −897 − −507 ms (R > 0.04) before the onset of SFW-1.

Averaged across the −880 − −485 ms interval before the onset of the SFW (corresponding to 120–515 ms after the onset of SFW-1), we found that the spatial pattern of neural activity was more similar in sentence pairs that predicted the same SFW (*within-pairs*: R = 0.074 + /- 0.02) than in pairs that predicted different SFWs (*between-pair*s: R = 0.067 + /- 0.02): $t_{(25)}$ = 3.751, p < 0.001, see *Figure 2B*. *Figure 2C* shows a scatter plot of the averaged R-values per participant within this interval. Eighteen out of 26 subjects had R-values below the diagonal, that is larger values for the *within-pair* than the *between-pair* spatial correlations. In contrast, there was no difference between the *within-pair* (R = 0.066 + /- 0.02) and *between-pair* (R = 0.068 + /- 0.02) spatial correlation values averaged across the −897 − −507 ms interval before the SFW-1 (corresponding to 103–493 ms after the onset of SFW-2): $t_{(25)}$ = −0.937, p = 0.358.

This difference in spatial similarity prior to the SFW cannot be explained by differences in the number of *within-pair* and *between-pair* sentences used to compute these mean spatial correlation values. This is because the number of trials per condition can affect the variance of the estimated mean value, but not the value of the estimated mean itself. Given that we carried out statistical analyses on the estimated mean values, the different number of *within-pairs* and *between-pairs* should not affect statistical inference at the participant level (*Groppe et al., 2011* and *Thomas et al., 2004*). Nonetheless, to convince skeptics, we repeated the analysis using a randomly selected subset of *between-pair* correlations that matched the number of *within-pair* correlations. This analysis confirmed that the *within-pair* spatial correlation values remained significantly greater than the *between-pair* correlations ($t_{(25)}$ = 2.393, p = 0.025; *Figure 2—figure supplement1*).

The difference in spatial similarity prior to the SFW cannot be explained by differences in lexical processing of the word before the SFW (SFW-1): this word always differed within sentence pairs, and any differences in its lexical properties (visual complexity, word frequency and syntactic class) between members of a pair were matched between pairs that constrained for the same SFW (*within-pairs*) and pairs that constrained for a different SFW (*between-pairs*). The spatial similarity effect also cannot be explained by differences in the predictability of the SFW-1: the cloze probability of these words was fairly low (11% on average) and any difference in cloze probability between members of a pair was matched between pairs that constrained for the same SFW (*within-pairs*) and pairs that constrained for a different SFW (*between-pairs*).

However, to fully exclude the possibility that the spatial similarity effect was driven by lexical processing of the SFW-1 rather than anticipatory processing of the SFW itself, we carried out an additional control analysis. First, we selected a subset of 31 pairs of sentences that contained exactly the same SFW-1, but nonetheless predicted a different SFW (43 unique SFWs; this *between-pairs* subset can be found in *Figure 1—source data 1*). Then we selected sentence pairs that constrained for these same SFWs (*within-pairs*), but which differed in the SFW-1. These constituted 43 *within-pair* sentences (also shown in *Figure 1—source data 1*). Various global contextual properties (such as

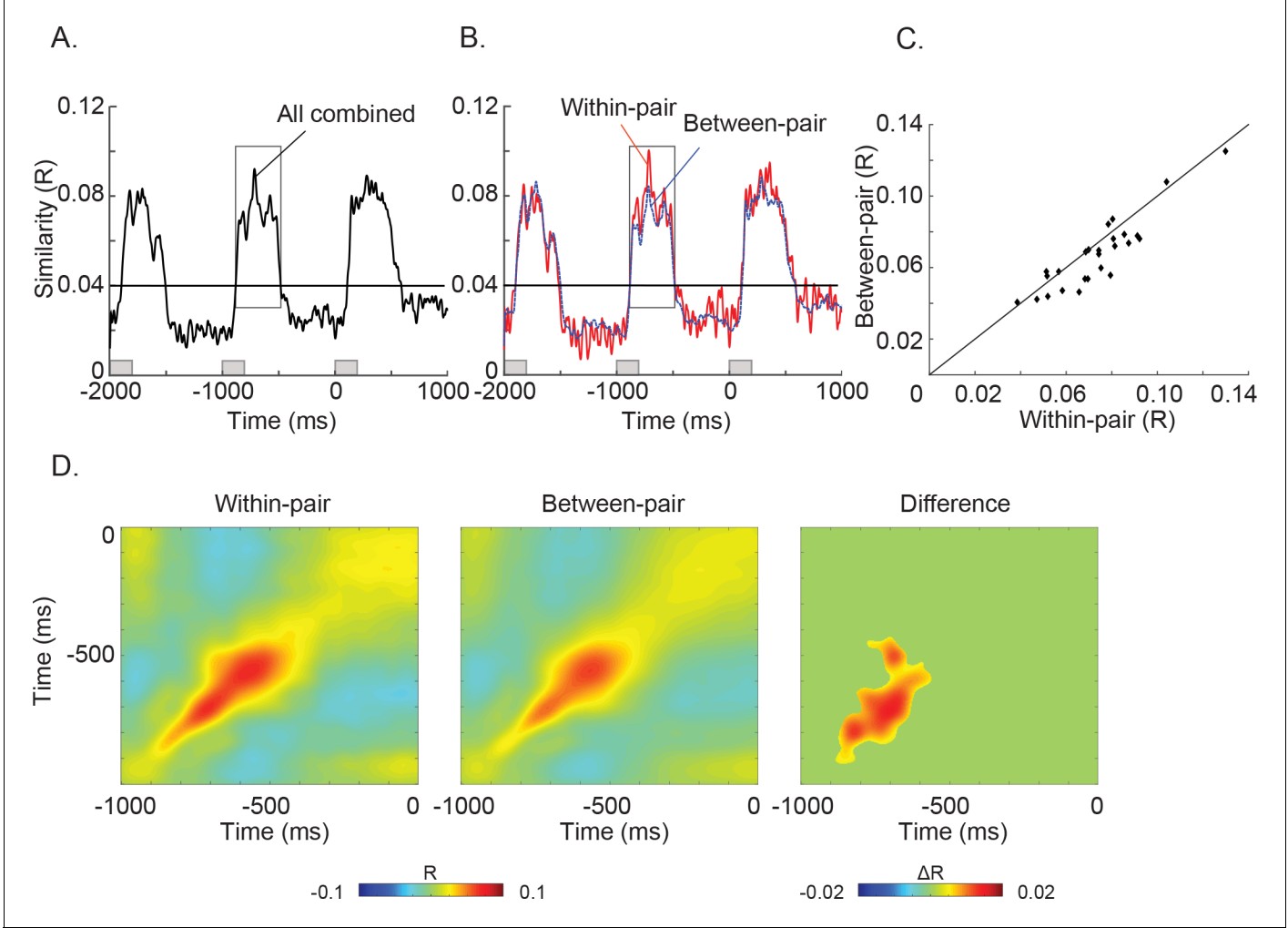

**Figure 2.** Results of the Spatial Representational Similarity Analysis. (**A**) The time series of spatial similarity R values combined across the *within-pair* and *between-pair* correlations. The horizontal line indicates a threshold of R = 0.04 where the general increase in spatial correlation was largest. (**B**) The time series of spatial similarity R values for pairs in which the same word was predicted (*within-pairs*, shown in red) and in which a different word was predicted (*between-pairs*, shown in blue). Both the *within-* and the *between-pair* spatial similarity time series showed a sharp increase at ~100 ms and a decrease at ~500 ms after the onset of each word. Between −880 and −485 ms before the onset of the final word, the spatial similarity was greater when the same word was predicted than when different words were predicted (*within-pairs* >*between-pairs*: $t_{(25)}$ = 3.751, p < 0.001). (**C**) Scatter plots of spatial similarity values averaged between −880 and −485 ms before the onset of the final word in 26 participants. In most participants (18/26) the *within-pair* spatial correlations were greater than the *between-pair* spatial correlations. (**D**) Cross-temporal spatial similarity matrices for the *within-* and *between-pair* correlations (Red: positive correlations; blue: negative correlations). Left and middle: Both sets of pairs showed increased spatial similarity along the diagonal with greater similarities for the *within-* than the *between-pairs* in the −900 − −500 ms interval prior to the onset of the final word. Right: The matrix shows the cluster with a statistically significant difference between the *within-pair* and *between-pair* spatial correlations (p = 0.002, cluster-randomization approach controlling for multiple comparisons over time). The absence of 'off-diagonal' correlations suggests that the spatial pattern of neural activity associated with the predicted word was reliable but changed over time.

DOI: https://doi.org/10.7554/eLife.39061.004

The following source data and figure supplements are available for figure 2:

**Source data 1.** Data used for plotting *Figure 2* as well as its supplementary Figures.

DOI: https://doi.org/10.7554/eLife.39061.009

**Figure supplement 1.** Results of the Spatial Representational Similarity Analysis after matching the number of pairs between the *within-pair* and *between-pair* correlations.

DOI: https://doi.org/10.7554/eLife.39061.005

**Figure supplement 2.** Results of the Spatial Representational Similarity Analysis in a subset of sentence pairs that had the same pre-sentence-final word (SFW-1) but predicted a different SFW (a subset of *between-pairs*, shown in blue), and a subset of sentences that constrained for these same SFWs, but which differed in the SFW-1 (a subset of *within-pairs*, shown in red).

*Figure 2 continued on next page*

*Figure 2 continued*

DOI: https://doi.org/10.7554/eLife.39061.006

**Figure supplement 3.** Results of the Spatial Representational Similarity Analysis for two subsets of trials where (**A**) sentences ending with expected words were seen first or (**B**) sentences ending with unexpected words were seen first.

DOI: https://doi.org/10.7554/eLife.39061.007

**Figure supplement 4.** Results of the Spatial Representational Similarity Analysis for pairs in which the same word was predicted (*within-pair*, shown in red) and in which the same syntactic category (e.g. nouns or verbs) of words (but not the same words) was predicted (*within-category*, shown in cyan).

DOI: https://doi.org/10.7554/eLife.39061.008

the number of words, number of clauses, and syntactic complexity) as well as the cloze probability of the SFW-1 were matched between this *within-pair* and *between-pair* subset (all ps > 0.05). We then compared the spatial similarity between these two subsets of sentence pairs. If the increased spatial similarity associated with the *within-pairs* versus *between-pairs* was due to the lexical processing of the SFW-1, then the spatial similarity should be greater in sentence pairs containing exactly the same SFW-1 (i.e. in the subset of *between-pairs*) than in sentence pairs that predicted the same SFW (i.e. in the subset of *within-pairs*). We found no evidence for this. Instead, we found that the averaged spatial similarity across the $-880 - -485$ ms interval before the onset of the SFW (corresponding to 120–515 ms after the onset of SFW-1) remained larger for the *within-pair* sentences (R = 0.072 + /- 0.02) than the *between-pair* sentences (R = 0.063 + /- 0.03): $t_{(25)}$ = 1.81, p = 0.08 (*Figure 2—figure supplement 2*), although in this subset analysis, the difference only approached significance due to the limited statistical power (on average there were only 40 *within-pairs* and 29 *between-pairs* after artifact rejection). Interestingly, the spatial correlation values, averaged across the $-897 - -507$ ms interval before the SFW-1 (corresponding to 103–493 ms after the onset of SFW-2), was larger for the *between-pairs* (R = 0.069 + /- 0.03) than the *within-pairs* (R = 0.058 + /- 0.03): $t_{(25)}$ = −2.295, p = 0.03. It is possible that this difference was driven by the prediction of the same SFW-1 in the *between-pairs*. However, this interpretation is speculative.

During sentence presentation, we avoided the repetition of the SFW (e.g. 'baby') within pairs by replacing the predicted SFW of one member of a pair with an unpredicted but plausible word in the other member of the pair (e.g. 'child'). However, one might argue that, after encountering the predicted word ('baby'), participants retained this item within memory, and that the increased spatial similarity of brain activity when reading the other member of the pair was due to anticipatory retrieval of this item that was facilitated by its previous presentation as a SFW. To address this concern, we divided the sentence pairs into two subsets according to whether the sentences with *expected* or *unexpected* SFWs were presented first. We then applied the spatial similarity analysis to both subsets (*Figure 2—figure supplement 3*) and compared their spatial similarity values. A repeated measures ANOVA with the factors Order (Expected SFW first, Unexpected SFW first) and Pairs (Within-pair, Between-pair) showed no main effect of Order ($F_{(1,25)}$ = 0.747, p = 0.396, $\eta^2$ = 0.029), nor an interaction between Order and Pairs ($F_{(1,25)}$ = 1.804, p = 0.191, $\eta^2$ = 0.067). We conclude that previously encountering a sentence ending with the expected SFW did not inflate the spatial similarity between sentence pairs that predicted the same SFW.

We then asked whether the increased spatial similarity associated with the *within-pair* versus *between-pair* sentences reflected the prediction of semantic features over and above the prediction of a general syntactic category (it is known that nouns and verbs are associated with distinct spatial patterns of activity; *Vigliocco et al., 2011*). To do this, we calculated *within-category* spatial similarity values by averaging the spatial similarity between all pairs of sentences that predicted the same syntactic category of words (i.e. nouns or verbs) and compared these values to the *within-pair* spatial similarity values. We found that the spatial similarity associated with pairs that predicted the same specific words (*within-pair* spatial similarity: R = 0.074 + /- 0.02) was significantly larger than the spatial similarity associated with pairs that predicted the same category (*within-category* spatial similarity: R = 0.068 + /- 0.02), ($t_{(25)}$ = −3.559, p = 0.002; *Figure 2—figure supplement 4*). This suggests that the greater *within-pair* versus *between-pair* spatial similarity effect was not simply reducible to the prediction of general syntactic category.

Finally, to further characterize the time course of brain activity reflecting unique lexico-semantic predictions, we correlated the spatial pattern of activity (across sensors) between each sentence (e.g. S1-A) at each time sample (e.g. $t_1$) with that of its paired sentence (e.g. S2-A') at all time

samples (e.g. from $t_1$ to $t_n$) in each participant (see also *King and Dehaene, 2014* and *Stokes et al., 2015a*), yielding a cross-temporal *within-pair* similarity matrix. We also calculated *between-pair* cross-temporal similarity matrices and averaged these within each participant and then across participants (*Figure 2D*). As expected, both the *within-* and *between-pair* group-averaged cross-temporal spatial similarity matrices showed that the spatial similarity was strongest around the diagonal in the first 500 ms after the onset of SFW-1. This was also the case for the difference between the *within-pair* and *between-pair* matrices (cluster-based permutation test: p = 0.002). This effect along the diagonal is consistent with the spatial similarity difference reported in *Figure 2B and C*. Moreover, the absence of an effect off the diagonal suggests that the spatial patterns associated with prediction changed over time.

## Temporal RSA: The temporal pattern of neural activity was more similar in sentence pairs that predicted the same versus different words, and this effect localized to left inferior temporal regions

As described above, across the $-880 - -485$ ms interval prior to the onset of the SFW, we observed a general increase in spatial similarity between all pairs of sentences, regardless of whether they constrained for the same SFW (*within-pairs*) or a different SFW (*between-pairs*). We next asked whether, within this time window, there were any brain regions in which the *temporal* pattern of neural activity was more similar for *within-pairs* than *between-pairs*. Note that this temporal RSA approach is fairly conservative in that it was limited to the time window that showed a spatial similarity effect, and so it may not have captured more extended temporal similarity effects that were not accompanied by a spatial similarity effect. The reason we took this approach is that we were interested, *a priori,* in any functional relationship between these measures, that is whether the spatial similarity effect reflected brain activity associated with the prediction of spatially distributed semantic representations, and whether the temporal similarity effect reflected brain activity associated with temporal binding of these spatially distributed representations. However, in order to fully exploit the spatiotemporal pattern of the data, future studies could examine the spatial and temporal patterns simultaneously using a spatiotemporal searchlight approach (*Nili et al., 2014*; *Su et al., 2012*; *Su et al., 2014*).

In each participant, we quantified the degree of temporal similarity of MEG activity produced by sentence pairs that predicted the same versus a different SFW by correlating the temporal pattern of signal produced within this time window at each sensor, yielding spatial topographic maps of temporal correlations. We then averaged these spatial temporal similarity maps within each participant and then across participants (*Figure 1C*). The group-averaged temporal similarity maps revealed a general increase in temporal similarity over bilateral temporal and posterior sensors, regardless of whether sentences predicted the same or a different SFW. When comparing the *within-* and *between-pair* temporal similarity topographic maps (*Figure 3A*), the temporal pattern of neural activity was more similar in pairs that predicted the same versus different words over central and posterior sensors (cluster-based randomization test: p = 0.002; *Figure 3A*: right panel). The comparison of a randomly selected subset of *between-pair* correlations that matched the number of *within-pair* correlations showed a similar, albeit slightly reduced, effect (marginally significant cluster: p = 0.0679; a cluster-randomization approach controlling for multiple comparisons over sensors; see *Figure 3—figure supplement 1A* for the group-averaged temporal similarity maps).

In order to estimate the underlying neuroanatomical source of the increased temporal similarity associated with the *within-pair* versus *between-pair* sentences, we repeated this analysis in source space. We first discretized the full brain volume using a grid. At each grid point, we constructed spatial filters using a 'beamforming approach' (a linearly constrained minimum variance technique; *Van Veen et al., 1997*) and applied it to the MEG data. Then, we performed the temporal similarity analysis on the time series from the spatial filters. The differences in the temporal similarity R-values were mapped on to the grid in each participant. These difference values were then morphed to the MNI brain and averaged. The source localization of the difference (corresponding to the difference of the topographic distribution, see *Figure 3A*: right panel) is shown in *Figure 3B*. It shows that the temporal pattern of neural activity was more similar in sentence pairs that predicted the same versus different words within a cluster over the left hemisphere (cluster-randomization controlling for multiple comparisons: p = 0.006). The strongest effect was found in the left inferior temporal lobe, as suggested by the 85% maximum difference of the temporal correlation values between the *within-pair* and *between-pair* sentences within the statistically significant cluster (see *Figure 3—figure*

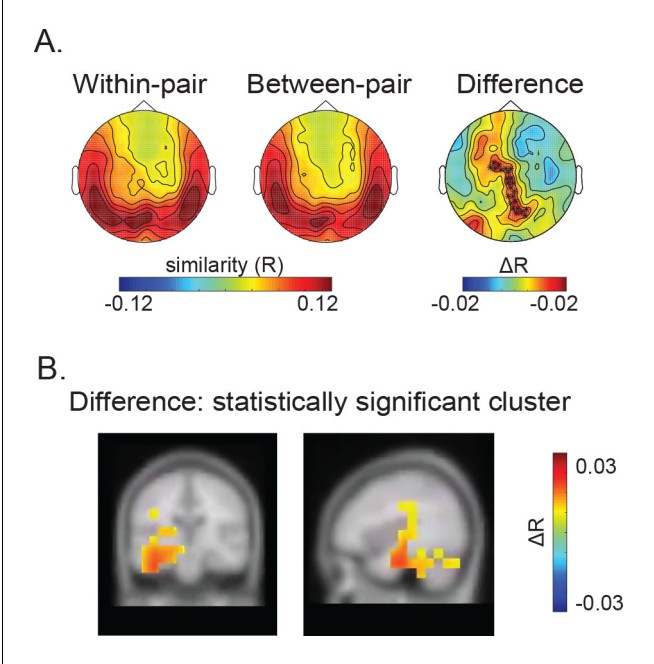

**Figure 3.** Results of the Temporal Representational Similarity Analysis. The Temporal Representational Similarity Analysis was carried out between −880 and −485 ms before the onset of the final word. (**A**) Temporal similarity topographic maps at the sensor level. Left and middle: Both the *within-* and *between-pair* correlations revealed increased temporal similarity over bilateral temporal and posterior sensors. Right: the difference map revealed greater temporal similarity when the same word was predicted (*within-pairs*) than when a different word was predicted (*between-pairs*) over central and posterior sensors. The sensors where this difference was significant at the cluster level are marked with black asterisks (p = 0.002; a cluster-randomization approach controlling for multiple comparisons over sensors). (**B**) Temporal similarity difference map in source space. The correlation values were interpolated on the MNI template brain and are shown both on the coronal plane (Talairach coordinate of peak: y = −19.5 mm) and the sagittal plane (Talairach coordinate of peak: x = −39.5 mm). This revealed significantly greater temporal similarity between sentence pairs that predicted the same word (*within-pairs*) than pairs that predicted a different word (*between-pairs*) within the left inferior temporal gyrus, extending into the medial temporal lobe including the left fusiform, hippocampus and parahippocampus as well as left cerebellum (p = 0.006; a cluster-randomization approach controlling for multiple comparisons over grid points).
DOI: https://doi.org/10.7554/eLife.39061.010

The following source data and figure supplements are available for figure 3:

**Source data 1.** Data used for plotting *Figure 3* as well as its supplementary Figures.
DOI: https://doi.org/10.7554/eLife.39061.013
**Figure supplement 1.** Results of the Temporal Representational Similarity Analysis after matching the number of pairs between the *within-pair* and *between-pair* correlations.
DOI: https://doi.org/10.7554/eLife.39061.011
**Figure supplement 2.** Results of the Temporal Representational Similarity Analysis showing the 85% maximum difference of the statistically significant cluster in source space.
DOI: https://doi.org/10.7554/eLife.39061.012

*supplement 2A*). The source extended medially into the left fusiform cortex, parahippocampus and hippocampus, as well as posteriorly into the left cerebellum. The comparison of a randomly selected subset of *between-pair* correlations that matched the number of *within-pair* correlations confirmed this finding (cluster-randomization controlling for multiple comparisons: p = 0.034, see *Figure 3—figure supplement 1B* for the statistically significant cluster and *Figure 3—figure supplement 2B* for the 85% maximum difference).

## Discussion

We asked whether the prediction of distinct words in highly constraining contexts is associated with distinct spatial and temporal patterns of neural activity before the appearance of new bottom-up input. To this end, we used MEG in conjunction with an RSA approach to index brain activity as participants read sentences in which the final word was highly predictable from the context. Based on a spatial correlation measure, we were able to provide evidence that the prediction of specific individual words produced unique spatial patterns of brain activity. This activity was evident between 120 and 515 ms following the word prior to the predicted sentence-final word (SFW-1). Moreover, within this time window, using a temporal correlation measure, we show that the prediction of specific individual words produced distinct temporal patterns of neural activity, which localized to the left inferior temporal region and neighboring areas. To the best of our knowledge, this is the first study to show that unique spatial and temporal patterns of neural activity are associated with the prediction of distinct words during language processing.

### Unique spatial patterns of neural activity are associated with the prediction of specific words, prior to the appearance of new bottom-up input

We found that the spatial pattern of neural activity (across sensors) was more similar in sentence pairs that predicted the same SFW (*within-pairs*) than in pairs that predicted a different SFW (*between-pairs*). This spatial similarity effect began at around 120 ms after the onset of the word before the SFW (i.e. the SFW-1). We interpret this finding as reflecting the greater spatial similarity in the pattern of brain activity produced by the prediction of the same word than the prediction of a different word. Before discussing this interpretation further, we first consider alternative possibilities.

One set of alternative interpretations is that, instead of reflecting similarities between the predicted SFW itself, the *within-* versus *between-pair* spatial similarity effect reflected a greater similarity in the pattern of brain activity evoked by the *contexts* of the sentence pairs that predicted the same word (*within-pairs*) versus a different word (*between-pairs*). The fact that the spatial similarity effect only became apparent after the onset of the word immediately preceding the SFW (i.e. after the SFW-1) suggests that it is unlikely to have reflected any *general* differences in these contexts (the full set of words prior to the SFW). Indeed, as noted in the Materials and methods, the two contexts within each pair were composed of distinct words, and any differences between members of a given pair in length (number of words) and complexity (number of clauses and syntactic complexity) were matched between pairs that constrained for the same SFW (*within-pairs*) and pairs that constrained for a different SFW (*between-pairs*). It is also unlikely that the spatial similarity effect reflected lexical differences of the SFW-1 itself because this always differed within pairs, and any differences in the visual complexity, frequency or syntactic class of the SFW-1 between the members of a given pair were again matched between pairs that constrained for the same word (*within-pairs*) and pairs that constrained for a different word (*between-pairs*). Finally, it is unlikely the spatial similarity effect reflected differences in the predictability of the SFW-1: the cloze probability of the SFW-1 was low (11% on average) and any difference between the members of a given pair in the cloze probability of the SFW-1 did not differ between pairs that constrained for the same SFW (*within-pairs*) and pairs that constrained for a different SFW (*between-pairs*).

Nonetheless, to rule out any possibility that the observed effect was driven by the lexical properties of the SFW-1, rather than the prediction of the SFW itself, we carried out an additional control analysis in a subset of sentence pairs that had the same SFW-1 but that predicted a different SFW (a subset of the *between-pair* sentences) and a subset of sentence pairs that constrained for these same SFWs, but that differed in the SFW-1 (a subset of the *within-pair* sentences). If the *within-pair* versus *between-pair* spatial similarity effect was driven by lexical similarities of the SFW-1, then we should have seen greater spatial similarity in pairs that contained exactly the same SFW-1, even though they constrained for a different SFW, than in pairs that contained a different SFW-1, even though they constrained for the same SFW. We found no evidence for this. Indeed, just as in our main analysis, the spatial patterns produced by the sentence pairs that predicted the same SFW (i.e. *within-pairs*) appeared to be more similar than the sentence pairs that predicted a different SFW (i.e. *between-pairs*), even though the *between-pairs* contained the same SFW-1. This strongly suggests

that the observed effect reflects the prediction of the SFW rather than lexical processing of the SFW-1.

A second set of alternative interpretations might acknowledge that the increase in spatial similarity detected in the *within-pair* sentences reflects activity related to the prediction of a specific SFW. However, instead of attributing the effect to the predicted representation itself, they might attribute it to participants' *recognition* of a match between the word that they had just predicted and a word that they had actually seen as the SFW earlier in the experiment. This seems unlikely because we found that the spatial similarity effect was just as large when the unexpected SFW of a pair was presented before the expected SFW, as when the expected SFW was presented first (see *Figure 2—figure supplement 3*). It is, however, conceivable that participants recognized a match between the word that they had just predicted and a word that they had predicted earlier in the experiment (even though this predicted word was never observed). For example, there is some evidence that a predicted SFW can linger in memory across four subsequent sentences, even if it is not actually presented (*Rommers and Federmeier, 2018*). This seems less likely to have occurred in the present study, however, where each member of a sentence pair was separated by at least 30 (on average 88) other sentences.

Our favored interpretation of the greater spatial similarity in brain activity produced by sentence pairs that constrained for the same word (*within-pairs*) versus a different word (*between-pairs*) is that it reflected activity associated with predicted SFW itself. This is by no means the first study to show evidence of lexico-semantic prediction before the onset of new bottom-up input during sentence processing. Several previous studies have reported evidence of such anticipatory processing following constraining relative to non-constraining contexts (see *Kuperberg and Jaeger, 2016a*, section 3.1), at least under experimental conditions that encourage the generation of high-certainty lexico-semantic predictions. What distinguishes the present study from this previous work is that it provides neural evidence that these lexico-semantic predictions are *item-specific* – that is, different predicted words are associated with spatially distinct patterns of neural activity.

This raises the question of exactly what type and grain of lexical information was reflected in these item-specific spatial patterns. In theory, an increased spatial similarity in association with sentence pairs that predicted the same upcoming word could have reflected greater similarity between the predicted word's syntactic, semantic, phonological, and/or its orthographic features. For example, a particular spatial pattern associated with a predicted word 'baby' could, in theory, reflect activity at the level of its syntactic category (e.g. <noun>), its lexico-semantic features (e.g. <human>, <small >, <cries>), its particular orthographic form (/b-a-b-y/) and/or its particular phonological form (e.g. /'beibi/).

We were able to exclude the possibility that our analysis simply picked up on syntactic category similarities between predicted words, such as whether they represented nouns or verbs, which are known to have distinct neuroanatomical representations (e.g. *Vigliocco et al., 2011*). This is because we designed our study such that 50% of the predicted SFWs were nouns and 50% were verbs, allowing us to calculate the *within-category* spatial similarity between all pairs of sentences that predicted the same syntactic category. We compared these *within-category* spatial similarity values with the *within-pair* spatial similarity values, in which the predicted SFWs shared *both* syntactic category and lexico-semantic features. We found that the *within-category* spatial similarity values were significantly smaller than the *within-pair* spatial similarity values (*Figure 2—figure supplement 4*). These findings suggest that the *within-pair* spatial similarity effect did not simply reflect the prediction of the broad syntactic category of the SFW.

Instead, we suggest that the spatial similarity effect reflected similarities at the level of the semantic properties and features that defined the meanings of the predicted words. As noted in the Introduction, the multimodal semantic properties and features associated with words are thought to be represented within regions that are spatially distributed across the cortex (*Damasio, 1989*; *Price, 2000*; *Martin and Chao, 2001*). We suggest that our analysis picked up distinct spatially distributed patterns of neural activity that corresponded to the particular sets of features associated with distinct predicted words. For example, the prediction of the particular set of semantic properties and features corresponding to the word <baby> (e.g. <human>, <small>, <cries>) may have been reflected by the activation of a particular spatially distributed network that differed from the network reflecting the prediction of the particular set of semantic features corresponding to a different predicted word, <roses> (e.g. <plant>, <scalloped petals>, <fragrant smell>).

It is also possible that the increased spatial similarity in association with sentence pairs that predicted the same word reflected similarities of predictions generated at a lower phonological and/or orthographic level of representation. On this account, the prediction of semantic features led to the top-down pre-activation of information at these lower levels of the linguistic hierarchy before new bottom-up information became available to these levels (see *Kuperberg and Jaeger, 2016a*, sections 3 and 5 for discussion). The present study cannot directly speak to this hypothesis. This is because, for the most part, there is a one-to-one correspondence between the semantic features and the phonological or orthographic forms of words. However, the methods described here provide one way of addressing this question in future studies. For example, by examining the spatial similarity of sentence pairs that constrain for words that share orthographic features but that differ in their meanings (homonyms), it should be possible to dissociate the prediction of orthographic/phonological representations from the prediction of semantic features associated with a given lexico-semantic item.

In addition to suggesting that the prediction of specific words is associated with unique spatial patterns of neural activity, our findings also provide some information about time course of such activity in relation to the appearance of new bottom input. As noted above, the spatial similarity effect began at 120 ms after the onset of the word before the predicted SFW (i.e. SFW-1), even though the effect was not driven by the lexical properties or the predictability of the SFW-1 itself. This provides evidence that the prediction of the SFW was generated at the first point in time at which participants had sufficient information to unambiguously generate this prediction. For example, in the sentence 'In the crib, there is a sleeping . . .', as comprehenders accessed the meaning of the word, <sleeping> , they may have also predicted the semantic features of <baby>. This type of account follows from a generative framework of language comprehension in which, following highly constraining contexts, comprehenders are able to predict entire events or states, along with their associated semantic features, prior to the appearance of new bottom-up input (e.g. *Kuperberg and Jaeger, 2016a*, sections 4 and 5; *Kuperberg, 2016b*; *St. John and McClelland, 1990*; *Rabovsky et al., 2018*). Importantly, however, we conceive of the spatial similarity effect detected here as primarily reflecting similarities at the level of semantic features (e. g. <human>, <small>, <crying>) associated with the predicted word ('baby'), rather than similarities of the entire predicted events/states (e.g. the <baby sleeping in the crib> event versus the <newborn baby in the hospital> event) (see *Kuperberg, 2016b*). As noted above, we cannot tell from the current findings whether these predicted semantic features, in turn, led to the top-down pre-activation of specific phonological or orthographic word-forms.

Despite its early onset, the spatial similarity effect was fairly short-lived: it lasted until 515 ms following the onset of the SFW-1 (corresponding to 315 ms following its offset at 200 ms), and then dropped off in the second half of the interval before the onset of the SFW (see *Figure 2B*). This was confirmed by the cross-temporal spatial similarity matrix (*Figure 2D*). The precise reason for this transient pattern is unclear. It is possible that, the predicted information was not maintained over the relatively long interstimulus interval used in the present study (SOA: 1000 ms per word). On the other hand, a failure to detect neural activity over a delay does not necessarily imply that this information is not present. This idea has been recently discussed in relation to the notion of 'activity-silent' working memory, which holds that representations within working memory can be maintained in a silent neural state, instead of being accompanied by persistent delayed activity (*Stokes, 2015b*; *Wolff et al., 2017*). Such content-specific silent activity can only be detected, if it is in the focus of attention and task-relevant. On this account, in the present study, despite the fact that we were not able to detect it, the predicted information was still present during the interstimulus interval, and it only became available once new bottom-up input was encountered. Of course, this interpretation is speculative, particularly given our use of a very slow presentation rate. It will be important for future work to determine whether similar dynamics are associated with the prediction of upcoming words when bottom-up inputs unfold at faster, more naturalistic rates.

Finally, we note that the cross-temporal spatial similarity matrix showed that the increased spatial similarity to the *within-pair* sentences was only found along the diagonal line, rather than generalizing across time points. This suggests that the unique spatial pattern of brain activity associated with the prediction of specific words changed over time. We speculate that this may be because different properties associated with particular words became available at different times. For example, the different semantic features (e.g. <human>, <small>, <cries>) associated with the prediction of a

specific word (e.g. 'baby') might have been recruited at different time points. As we discuss next, temporal binding may play a role in integrating these dynamically evolving spatial patterns of activity to instantiate specific lexico-semantic predictions.

## Unique temporal patterns of neural activity within the left inferior and medial temporal lobe are associated with the prediction of specific words

In addition to being associated with unique *spatial* patterns of neural activity, we also found evidence that the prediction of specific words was associated with unique *temporal* patterns of neural activity. Specifically, across the time window that showed the increased spatial similarity effect, a cluster of MEG sensors revealed a greater similarity in the temporal pattern of brain activity in pairs of sentences that predicted the same SFW (*within-pairs*) versus a different SFW (*between-pairs*). Moreover, we localized the source of this effect to the left ventral and medial temporal lobe.

This observation is in line with a recent study that applied temporal RSA to intracranial EEG signals, and reported that the temporal pattern of neural activity within the left inferior temporal lobe encoded item-specific representations during picture naming (*Chen et al., 2016*). The current findings extend these previous results by suggesting that temporal similarity patterns corresponding to unique lexico-semantic items can be detected *before* new bottom-up input becomes available.

The precise functional significance of the temporal similarity effect is unclear. However, we suggest that it is consistent with a classic theory by *Damasio (1989)*, who proposed that multimodal semantic features, represented in widely distributed regions of the cortex, become bound together through a process of 'temporal synchrony', and that this binding occurs within 'convergence zones', which act to unify these features into a discrete whole (*Damasio, 1989*). In the present study, it is possible that the unique temporal patterns of neural activity within the left ventral/medial temporal lobe played a functional role in binding the unique sets of semantic features that were represented by the unique spatial patterns of neural activity. Speculatively, these unique temporal patterns of neural activity may have also played a role in binding this information as it became available dynamically over time (as opposed to becoming available all at once). For example, by tracking (or perhaps even orchestrating) the particular time-course of accessing the distributed brain regions that represent the semantic features of <baby> (e.g. <human>, <small> and <cries>), a particular temporal signature may have functioned to bind the dynamically evolving and spatially distributed pattern of neural activity into a coherent lexico-semantic representation, corresponding to participants' subjective experiences of predicting a specific word.

The localization of the temporal similarity effect to the left ventral temporal regions (left inferior temporal lobe and fusiform cortex) is consistent with the well-established role of these regions in lexico-semantic processing (*Lüders et al., 1991*; *Lüders et al., 1986*; *McCarthy et al., 1995*; *Mummery et al., 1999*; *Nobre and McCarthy, 1995*; *Visser et al., 2010*). In particular, it is consistent with the proposed role of these regions as 'hubs' that brings together widely distributed semantic features across the cortex (*Patterson et al., 2007*; *Ralph et al., 2017*). Such hubs may function as a 'dictionary' by mediating between widely distributed conceptual knowledge and specific word forms (orthographic and phonological knowledge) (*Caramazza, 1996*; *Damasio et al., 1996*) and/or they play a more domain-general role in semantic processing (*Nobre et al., 1994*; *Patterson et al., 2007*; *Reddy and Kanwisher, 2006*; *Shimotake et al., 2015*). By showing that this region can encode unique temporal patterns of neural activity that correspond to unique lexico-semantic predictions, our findings shed further light on how these regions might actually instantiate this type of binding.

Notably, the activity within the inferior temporal cortex extended into the medial temporal lobe (the parahippocampal gyrus and the hippocampus). While MEG source-modeling results within medial and subcortical regions should be interpreted with caution, the possible involvement of the hippocampus is interesting given other work that has implicated it as playing a crucial role in binding representations to generate predictions. A large literature from recordings in rats demonstrates that the hippocampus represents upcoming spatial representations as the rat is navigating (*Gupta et al., 2012*), and we have a good understanding of the physiological mechanisms supporting such predictions (*Lisman and Redish, 2009*). There is also growing evidence that these predictive mechanisms might generalize to the human hippocampus (*Chen et al., 2011*; *Davachi and DuBrow, 2015*; *Harrison et al., 2006*; *Hindy et al., 2016*; *Schiffer et al., 2012*). Moreover, recently, it was found

that the temporal patterns in higher frequency bands recorded within the hippocampus were similar between a pre-picture interval and the picture itself (*Jafarpour et al., 2017*), suggesting a role in representing pre-activated non-verbal semantic information. Given these findings, it is conceivable that the hippocampus also plays an analogous role in language prediction. Indeed, *Piai et al. (2016)* used intracranial recordings in humans to demonstrate predictive effects in the hippocampus in a language task in which the sentence-final word had to be produced (*Piai et al., 2016*).

In addition to the medial temporal lobe, the activity also included the left cerebellum. Again, given the limited spatial resolution of MEG, this activation should be interpreted with caution. However, previous studies have reported bilateral cerebellum activation (right dominant) during language prediction (*Bonhage et al., 2015*; *Lesage et al., 2017*; *Wang et al., 2018*). Our findings seem to suggest that the cerebellum also may play a role in generating item-specific predictions in language processing.

Before concluding, we emphasize that this study does not speak to the debate about whether neural evidence of anticipatory processing, particularly at the level of specific word-forms (rather than semantic features), can be detected during sentence comprehension under conditions that do not encourage predictive processing (*DeLong et al., 2005*; *Nieuwland et al., 2018*; *Yan et al., 2017*); nor does it address the question of whether such predictions are probabilistic in nature. In the present study, we deliberately used highly constraining contexts to encourage participants to generate high certainty specific lexico-semantic predictions, and we used a long interstimulus interval between words to ensure that we would be able to detect any representationally specific neural activity if it was present. We see the unique contribution of our study as providing evidence that, when we know that item-specific lexico-semantic are generated, they are associated with unique spatial and temporal patterns of neural activity. These findings pave the way toward the use of these methods to determine whether and when such item-specific lexico-semantic representations become available as language, in both visual and auditory domains, unfolds more rapidly in real time.

## Conclusion

In conclusion, we used MEG to show that unique patterns of neural activity are associated with the prediction of specific lexico-semantic items during language processing. We showed that unique *spatial* patterns became active at around 100 ms after a word was unambiguously predicted and that their activation was transient and dynamic. In addition, we show that the prediction was accompanied by unique *temporal* patterns of brain activity that localized to the left inferior and medial temporal lobe.

## Materials and methods

### Design and development of stimuli

We developed a stimulus set of 120 pairs of sentences in Mandarin with highly constraining contexts. The two contexts within each pair were distinct from one another, and they had no content words in common (with the exception of five pairs), but they each strongly predicted the same sentence-final word (SFW). For example, in *Figure 1A*, both sentences S1 and S2 predicted the word, 'baby'. In half of these sentences, the expected final word was a noun and in the other half, it was a verb.

To select and characterize this final set of sentences, we began with an initial set of 208 pairs and carried out a cloze norming study in 30 participants (mean age: 23 years; range: 18–28 years old; 15 males), who did not participate in the subsequent MEG study. In this cloze study, sentence contexts were presented without the SFW (e.g. 'In the crib there is a sleeping …') and participants were asked to complete the unfinished sentence by writing down the most likely ending. The two members of each sentence pair were counterbalanced across two lists (with order randomized within lists), which were each seen by half the participants. Testing took approximately 40 min per participant.

To calculate the lexico-semantic constraint of each sentence context, we tallied the number of participants who produced the most common completion for a given context. We retained 66 pairs in which 73% of the participants predicted the same SFW, that is at least 11 out of 15 participants filled in the same word in each sentence pair. We then revised 103 sentences (54 sentences in list 1 and 49 in list 2) to make them more constraining, and we re-tested them in the same group of

participants. After this second round of cloze testing, we selected the final set of 120 sentences for the MEG experiment. In the final set of stimuli, the lexico-semantic constraints of 109 pairs were above 70% and the constraints of the remaining 11 pairs were slightly lower (mean: 58%; SD: 12). Across all pairs, the mean lexico-semantic constraint was 88% (SD: 12).

We then generated full sentences by adding a SFW to each member of a pair. In one member of each pair, this SFW was highly predictable; it was the most common word filled by the cloze participants (e.g. 'baby' following context S1, 'In the crib there is a sleeping...'). In the other member of the pair, we selected a word that was semantically related to the highly predicted word but was not produced by any of the participants in the cloze norming, with the whole sentence still being plausible (e.g. 'child' following context, S2, 'In the hospital, there is a newborn...'). Thus, for this sentence, the lexical cloze probability was zero, see *Figure 1A* for examples. All sentence contexts (e.g. S1 and S2) were combined with both lexically predicted (e.g. A: 'baby') and unpredicted (e.g. A': 'child') SFWs, for example S1-A, S1-A', S2-A, S2-A' and the SFWs were then counterbalanced across two lists, ensuring that, in the MEG session, each participant saw both members of each sentence pair, but no participant saw the same SFW twice. Within each list, sentences were pseudo-randomized so that participants did not encounter more than three expected or unexpected SFWs in succession, and the two members of each pair were presented apart from each other, with at least 30 (on average 88) sentences that predicted different words in between. All Mandarin sentences, together with their English translations, are available in the *Figure 1—source data 1*.

We measured a number of properties of the sentence contexts up until the SFWs and determined whether these properties differed systematically between pairs of contexts that predicted the same SFWs (i.e. *within-pairs*) and pairs of contexts that predicted different SFWs (i.e. *between-pairs*). We counted the number of words in each sentence context (ranging from 4 to 12 words), and the number of clauses within each sentence (ranging from 1 to 4 clauses). We also marked whether there was embedded dependency in each sentence. Then, for each possible pair of sentences, we categorized whether their contexts differed (marked as 1) or not (marked as 0) from one another on each of these three measures. These values were used as the dependent variable in independent sample t-tests (*within-pairs*; N = 120; *between-pairs*; N = 120*119*2 = 28560). The tests showed that any differences in the number of words, number of clauses, and syntactic complexity were matched between pairs that constrained for the same word (*within-pairs*) and pairs that constrained for a different word (*between-pairs*): all ps > 0.20.

We also examined several lexical properties of the SFW-1 to make sure that any observed spatial or temporal similarity effect could not be explained by lexical processing of the SFW-1 itself. The Chinese SFW-1s as well as their English translations can be found in *Figure 1A—source data 1*. We coded the syntactic class of the SFW-1 as either a content word (verb, noun, adjective, adverb) or a function word (pronoun, classifier, conjunction, particle, prepositional phrases) and marked whether the syntactic class of the SFW-1 differed (marked as 1) or not (marked as 0) within members of each possible pair of sentences. In Chinese, the SFW could be either a word or a phrase, each containing several characters (ranging from 1 to 5). We assessed the visual complexity of each SFW-1 by aggregating the number of strokes of all characters, and, for each possible pair of sentences, we calculated the absolute difference in the number of strokes of the SFW. We also extracted word frequency values for each SFW-1 (measured as the log10 transformed n-gram frequency out of one million) from *Sun, 2003* in 82% of the stimuli and from *Da (2004)* in 10% of the stimuli; the values of the remaining 8% of SFW-1s whose frequency did not appear in either database were marked as zero, and calculated the absolute difference of the word frequency values of the SFW-1 for each possible pair of sentences. These values were used as the dependent variable in independent sample t-tests (*within-pairs*; N = 120; *between-pairs*; N = 120*119*2 = 28560). The tests showed that any differences in the syntactic class, the visual complexity and the frequency of the SFW-1 were matched between pairs that constrained for the same word (*within-pairs*) and pairs that constrained for a different word (*between-pairs*), all ps > 0.40.

Finally, we assessed the cloze probability of the SFW-1 in a new group of 30 participants (mean age: 24 years; range: 19–28 years old; 15 males). In this test, sentence contexts were presented up until the SFW-2 (e.g. 'In the crib there is a ...'), and the 120 pairs of sentences were counterbalanced across two lists, which were each seen by 15 participants. The cloze probability of the SFW-1 was calculated by tallying the number of participants who produced the SFW-1 for a given context. Overall, the cloze of the SFW-1 was low (11.33% ± 20.25% on average). Once again, for each possible

pair of sentences, we calculated the absolute difference in the cloze probability of the SFW-1 and carried out an independent sample t-test. Once again, any differences in cloze probability were matched between pairs that constrained for the same word (*within-pairs:* 17.00% cloze difference) and pairs that constrained for a different word (*between-pairs*: 17.28% cloze difference), $t_{(28678)} = -0.136$, p = 0.89.

## Participants in the MEG study

The study was approved by the Institutional Review Board (IRB) of the Institute of Psychology, Chinese Academy of Sciences. Thirty-four students from the Beijing area were initially recruited by advertisement. They were all right-handed native Chinese speakers without histories of language or neurological impairments. All gave informed consent and were paid for their time. The data of eight participants were subsequently excluded because of technical problems, leaving a final MEG dataset of 26 participants (mean age 23 years, range 20–29; 13 males).

## Procedure

MEG data were collected while participants sat in a comfortable chair within a dimly-lit shielded room. Stimuli were presented on a projection screen in a grey color on a black background (visual angle ranging from 1.22 to 2.44 degrees). As shown in *Figure 1A*, each trial began with a blank screen (1600 ms), followed by each word with an SOA of 1000 ms (200 ms presentation with an inter-stimulus interval, ISI, of 800 ms). The final word ended with a period followed by a 2000 ms inter-trial interval. After one-sixth of the trials, at random, participants read either a correct or an incorrect statement that referred back to the semantic content of the sentence that they had just read (for example, S1-A and S2-A' in *Figure 1A* might be followed by the incorrect statement, 'There is an old man.'). Participants were instructed to judge whether or not the statements were correct by pressing one of two buttons with their left hand. This helped ensure that participants read the sentences for comprehension. In all other trials, the Chinese word '续 继' (meaning 'NEXT') appeared, and participants were instructed to simply press another button with their left hand within 5000 ms in order to proceed to the next trial.

The 240 sentences were divided into eight blocks, with each block lasting about 8 min. Between blocks there was a small break during which participants were told that they could relax and blink, but to keep the position of their heads still. Participants could start the next block by informing the experimenter verbally. The whole experiment lasted about 1.5 hr, including preparation, instructions and a short practice session consisting of eight sentences.

## MEG data acquisition

MEG data was collected using a CTF Omega System with 275 axial gradiometers at Institute of Biophysics, Chinese Academy of Sciences. Six sensors (MLF31, MRC41, MRF32, MRF56, MRT16, MRF24) were non-functional and were therefore excluded from the recordings. The ongoing MEG signals were low-pass filtered at 300 Hz and digitized at 1200 Hz. Head position, with respect to the sensor array, was monitored continuously with three coils placed at anatomical landmarks (fiducials) on the head (forehead, left and right cheekbones). The total movement across the whole experiment was, on average, 8 mm across all participants. In addition, structural Magnetic Resonance Images (MRIs) of 25 participants were obtained using a 3.0T Siemens system. During MRI scanning, markers were attached in the same position as the head coils, allowing for later alignment between these MRIs and the MEG coordinate system.

## MEG data processing

MEG data were analyzed using the Fieldtrip software package, an open-source Matlab toolbox (*Oostenveld et al., 2011*). In order to minimize environmental noise, we applied third order synthetic gradiometer correction during preprocessing. Then, the MEG data were segmented into 4000 ms epochs, time-locked from the onset of two words before the SFW (SFW-2) until 2000 ms after the onset of the SFW. Trials (i.e. whole epochs) contaminated with muscle or MEG jump artifacts were identified and removed using a semi-automatic routine. We then carried out an Independent Component Analysis (ICA; *Bell and Sejnowski, 1997*; *Jung et al., 2000*) and removed components associated with the eye-movement and cardiac activity from the MEG signal (about five components

per subject). Finally, we inspected the data visually and removed any remaining artifacts. On average, 96% ± 3.4% of trials were retained.

## Spatial Representational Similarity Analysis
### Calculation of spatial similarity time series

A schematic illustration of the spatial representational similarity analysis (RSA) approach is shown in Figure 1B. First, we detrended and applied a 30Hz low pass filter to the MEG data. Next, in each participant, for each trial, and at each time sample, we extracted a vector of MEG data that represented the spatial pattern of activity across all 269 MEG sensors (6 of 275 sensors were not operational). We then quantified the degree of spatial similarity of MEG activity produced by the two members of each sentence pair predicting the same SFW (e.g. between S1-A and S2-A', in Figure 1A) by correlating their spatial vectors at consecutive time samples across the 4000ms epoch. This yielded a time-series of correlations (Pearson's r values) reflecting the degree of spatial similarity at each time sample between sentences that predicted the same SFW (e.g. time-series $R^1_{within}$ and $R^2_{within}$, see Figure 1B). We refer to these as *within-pair* spatial similarity time series. After artifact rejection, in each participant, there were, on average, N = 111+/-8 complete *within-pair* spatial similarity time series. We then averaged these time series together to yield an average *within-pair* spatial similarity time series within each participant ($\frac{1}{N}\sum_{i=1}^{N} R^i_{within}$; *Figure 1B*).

We then repeated this entire procedure, but this time we correlated spatial patterns of MEG activity between pairs of sentences that predicted a different SFW, for example, between S1-A and S3-B (*Figure 1A*). This yielded 2N(N-1) *between-pair* spatial correlation time courses, for example $R^1_{between}$ and $R^2_{between}$ (*Figure 1B*). We again averaged these together to yield a time series of R-values within each participant ($\frac{1}{2N(N-1)}\sum_{i=1}^{2N(N-1)} R^i_{between}$; *Figure 1B*), which reflected the degree of similarity between spatial patterns of activity elicited by sentences that predicted different SFWs at each time sample (i.e. *between-pair* spatial similarity time series). *Figure 2B* shows the averages, across all participants, of the *within-pair* and the *between-pair* spatial similarity time series (see *Figure 2—source data 1*).

### Calculation of cross-temporal spatial similarity matrices

To characterize how temporally sustained the spatial patterns were (see also *King and Dehaene, 2014*; *Stokes et al., 2015a*), in each participant, for each sentence pair that constrained for the same SFW, we correlated the spatial pattern vector between one member of the pair (e.g. S1-A) at a particular time sample (e.g. $t_1$) with that of the other member (e.g. S2-A') at all time samples (e.g. from $t_1$ to $t_n$), thereby constructing cross-temporal similarity matrices for all *within-pair* sentences, with each entry representing the spatial similarity between two sentences at two time samples (e.g. $R_{(i,j)}$ represents the correlation between S1-A at time *i* and S2-A' at time *j*). In order to increase computational efficiency, we down-sampled the data to 300 Hz, and we smoothed the resulting correlation values in time with a Gaussian kernel (40 ms time window, SD: 8 ms). We then averaged the cross-temporal similarity matrices across all *within-pair* sentences within each participant and then across participants (*Figure 2D*: left). The R-values along the diagonal reflect the spatial similarity at corresponding time samples ($R_{(i,j)}$ when i = j; that is the time series of similarity R-values as described in *Figure 1B*), while the R-values off the diagonal reflects cross-temporal spatial similarity. We then repeated this entire procedure for pairs of sentences that predicted a different SFW (*between-pairs*). We randomly selected N *between-pairs* to match with the N *within-pairs* for averaging in order to increase computational efficiency (*Figure 2D*: middle). *Figure 2D* (right) shows the difference for the group-averaged *within-pair* and *between-pair* cross-temporal spatial similarity matrices.

### Statistical testing

As can be seen in *Figure 2A*, the averaged *within-pair* and the *between-pair* spatial similarity time series showed a sharp increase in R-values at around 100 ms after the onset of the word before the SFW (SFW-1) lasting for about 400 ms (i.e. 300 ms into the ISI after the SFW-1 offset) before sharply decreasing again. This pattern of a sharp increase and decrease in spatial correlations was also seen in association with the previous word (SFW-2) as well as the following word (SFW). In order to

objectively quantify the time-window over which this general increase in spatial similarity R values was sustained during the prediction period, we compared the averaged *within-pair* and *between-pair* spatial similarity time series against a threshold of R = 0.04 based on visual inspection of the R-values in the prediction time window. We found an increase in R-values from −880 ms to −485 ms (i.e. 120 ms to 515 ms relative to the onset of SFW-1), as well as from −1897 to −1507 ms before the onset of the SFW (i.e. 103 ms to 493 ms relative to the onset of SFW-2) (*Figure 2B*). Similar results were found for a threshold of R = 0.03.

We then averaged across the −880 – −485 ms interval before the onset of the SFW and carried out paired t-tests to determine whether, collapsed across this time window, the spatial pattern of MEG activity produced by sentence pairs that predicted the same SFW was significantly more similar than the spatial pattern of MEG activity produced by sentences that predicted different SFW (i.e. *within-pair* vs. *between-pair* spatial correlation R values). We repeated the same analysis for the −1897 – −1507 ms interval before the onset of the SFW (i.e. −897 – −507 ms before the onset of the SFW-1).

To test for cross-temporal statistical differences in spatial similarity patterns produced by sentence pairs that predicted the same versus different SFWs while controlling for multiple comparisons over time, we applied a cluster-based permutation approach (*Maris and Oostenveld, 2007*): We first carried out paired t-tests at each data time sample in the cross-temporal spatial similarity matrices within the 1000 ms interval between the onset of SFW-1 and the onset of SFW. Data points that exceeded a pre-set uncorrected p-value of 0.05 or less were considered temporal clusters. The individual t-statistics within each cluster were summed to yield a cluster-level test statistic — the cluster mass statistic. We then randomly re-assigned the spatial similarity R values across the two conditions (i.e. *within-pair* and *between-pair*) at each data point within the matrix, within each participant, and calculated cluster-level statistics as described above. This was repeated 1000 times. For each randomization, we took the largest cluster mass statistic (i.e. the summed T values), and, in this way, created a null distribution for the cluster mass statistic. We then compared our observed cluster-level test statistic against this null distribution. Any temporal clusters falling within the highest or lowest 2.5% of the distribution were considered significant.

## Temporal Representational Similarity Analysis
### Construction of temporal similarity maps at sensor level

In each participant, at each sensor for each trial, we considered the MEG time series in the −880 – −485 ms interval before the onset of the SFW — that is, the time-window over which we observed the general increase in spatial similarity R values during the prediction period (see *Figure 2A*). At each sensor, we then correlated this time series within this window between the two members of each sentence pair that predicted the same SFW (e.g. between S1-A and S2-A', in *Figure 1A*) to yield an R value representing the degree of temporal similarity: an R value of 1 implies that the two time series are in perfect synchrony; an R value of 0 implies that the two time series are not correlated, while an R value of −1 implies that the two time series are anti-correlated. Together, these R values at each sensor yielded *within-pair* temporal similarity topographic maps for each pair, for example topographic maps $R^1_{within}$ and $R^2_{within}$, see *Figure 1C*. We then averaged across all the *within-pair* temporal correlations at each sensor to yield an average *within-pair* temporal similarity topographic map within each participant and then averaged across participants (see *Figure 1C*).

We then repeated this procedure, but this time correlating time series from MEG sensors produced by pairs of sentences that predicted a different SFW, yielding topographic maps of the *between-pair* temporal correlations (e.g. $R^1_{between}$ and $R^2_{between}$ in *Figure 1C*). These maps were again averaged together to yield an average topographic map of R values within each participant, and then averaged across participants (*Figure 1C*).

### Construction of temporal similarity maps at source level

We also constructed temporal similarity maps at the source level. We estimated the MEG signals at the source level by applying a spatial filter at each grid point using a beamforming approach (*Van Veen et al., 1997*). We computed a linearly constrained minimum variance (LCMV; *Van Veen et al., 1997*) spatial filter on the 30 Hz low-pass filtered (and linearly detrended) data from onset of SFW-1 to 1000 ms after onset of SFW (i.e. −1000–1000 ms relative to SFW onset). The LCMV

approach estimates a spatial filter from a lead field matrix and the covariance matrix of the data from the axial gradiometers. To obtain the lead field for each participant, we first spatially co-registered the individual anatomical MRIs to the sensor MEG data by identifying the fiducials at the forehead and the two cheekbones. Then a realistically shaped single-shell head model was constructed based on the segmented anatomical MRI for each participant (*Nolte, 2003*). Each brain volume was divided into a grid with 10 mm spacing and the lead field was calculated for each grid point. Then the grid was warped on to the template Montreal Neurological Institute (MNI) brain (Montreal, Quebec, Canada). The MNI template brain was used for one participant whose MRI image was not available. The application of the LCMV spatial filter to the sensor-level data resulted in single-trial estimates of time series at each grid point in three orthogonal orientations. To obtain one signal per grid point we projected the time series along the direction that explains most variance using singular value decomposition. In order to construct temporal similarity maps in source space, we followed the same procedures as above, by correlating the time series at each grid point. The grand-average similarity values were interpolated onto the MNI template brain (*Figure 3B*; see *Figure 3—source data 1*).

## Testing for significant differences between the within- and between-pair temporal similarity maps

To compare the *within-pair* vs. *between-pair* temporal correlation R values statistically, both at the sensor level and at the source level, we took a cluster-based permutation approach, controlling for multiple comparisons over sensors or grid-points (*Maris and Oostenveld, 2007*). At each sensor/grid point, in each participant, we compared the mean differences in the temporal similarity R values between sentence pairs predicting the same word (i.e. *within-pair*) versus a different word (i.e. *between-pair*). Sensors within 40 mm that exceeded the 95th percentile of the mean difference were considered clusters. We used the mean difference for thresholding the clusters in order to account for the overall R-value difference across participants. In source space, clusters were formed by contiguous grids points. Within each cluster, we then summed the mean differences of R values at each sensor/grid-point to yield a cluster-level test statistic — the cluster mass statistic. Next, we randomly re-assigned the R-values across the two conditions (i.e. *within-pair* and *between-pair*) at each sensor/grid within each participant, and calculated cluster-level statistics as described above. This was repeated 1000 times. For each randomization, we considered the largest cluster mass statistic (i.e. the summed mean difference within a cluster to create a null distribution for the cluster mass statistic). Then we compared our observed cluster-level test statistic against this null distribution. Any clusters falling within the highest or lowest 2.5% of the distribution were considered significant.

## Acknowledgements

This work was funded by the Natural Science Foundation of China (31540079 to LW), the National Institute of Child Health and Human Development (R01 HD08252 to GRK), and a James S McDonnell Foundation Understanding Human Cognition Collaborative Award (220020448), Wellcome Trust Investigator Award in Science (207550) and the Royal Society Wolfson Research Merit Award to OJ. It was supported in part by the Ministry of Science and Technology of China grants (2012CB825500, 2015CB351701). We thank Yang Cao and Yinan Hu for their assistance with data collection.

## Additional information

### Funding

| Funder | Grant reference number | Author |
| --- | --- | --- |
| National Natural Science Foundation of China | 31540079 | Lin Wang |
| Ministry of Science and Technology of the People's Republic of China | 2012CB825500 | Lin Wang |

| Ministry of Science and Technology of the People's Republic of China | 2015CB351701 | Lin Wang |
|---|---|---|
| National Institute of Child Health and Human Development | R01 HD08252 | Gina Kuperberg |
| James S. McDonnell Foundation | Understanding Human Cognition Collaborative Award: 220020448 | Ole Jensen |
| Wellcome Trust | Investigator Award in Science: 207550 | Ole Jensen |
| Royal Society | Wolfson Research Merit | Ole Jensen |

The funders had no role in study design, data collection and interpretation, or the decision to submit the work for publication.

## Author contributions

Lin Wang, Conceptualization, Data curation, Software, Formal analysis, Funding acquisition, Investigation, Visualization, Methodology, Writing—original draft, Project administration, Writing—review and editing; Gina Kuperberg, Conceptualization, Supervision, Funding acquisition, Visualization, Writing—original draft, Project administration, Writing—review and editing; Ole Jensen, Conceptualization, Software, Supervision, Funding acquisition, Visualization, Methodology, Writing—original draft, Project administration, Writing—review and editing

## Author ORCIDs

Lin Wang http://orcid.org/0000-0001-6911-0660
Gina Kuperberg http://orcid.org/0000-0001-6093-7872
Ole Jensen http://orcid.org/0000-0001-8193-8348

## Ethics

Human subjects: The study was approved by the Institutional Review Board (IRB) of the Institute of Psychology, Chinese Academy of Sciences (H15037). Thirty-four students from the Beijing area were initially recruited by advertisement. All gave informed consent and were paid for their time.

## Decision letter and Author response

Decision letter https://doi.org/10.7554/eLife.39061.018
Author response https://doi.org/10.7554/eLife.39061.019

# Additional files

## Supplementary files

• Source code 1. Matlab code for data analysis. These scripts were used for spatial Representational Similarity Analysis, cross-temporal Spatial Similarity Analysis, as well as temporal Representational Similarity Analysis.
DOI: https://doi.org/10.7554/eLife.39061.014

• Transparent reporting form
DOI: https://doi.org/10.7554/eLife.39061.015

## Data availability

All data generated or analysed during this study are included in the manuscript and supporting files. Source data files have been provided for Figures 2 and 3.

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
