## [Decision Letter]

Thank you for submitting your article "Specific lexico-semantic predictions are associated with unique spatial and temporal patterns of neural activity" for consideration by *eLife*. Your article has been reviewed by three peer reviewers, and the evaluation has been overseen by a Reviewing Editor and Joshua Gold as the Senior Editor. The reviewers have opted to remain anonymous.

The reviewers have discussed the reviews with one another and the Reviewing Editor has drafted this decision to help you prepare a revised submission.

Summary:

The manuscript describes an MEG study which shows greater spatial and temporal similarity of neural responses prior to the onset of an identical target word (e.g. "baby") in two sentences that share the same predicted final word ("In the crib, there is a sleeping…" vs "In the hospital, there is a newborn…") compared to equivalent responses in entirely unrelated sentences.

This greater similarity is seen over an extended time period during and after presentation of the penultimate word of the sentence ("sleeping" and "newborn"), though neural similarity is no longer apparent for around 400ms prior to the onset of the sentence final word, which was either as predicted ("baby") or a synonym ("child"). Source localisation of sensors showing maximum similarity localises this effect to an extended set of left inferior and medial temporal regions, "consistent with the well-established role of these regions in lexico-semantic processing". Results are interpreted as providing evidence in favour of "pre-activation of distinct words during language processing" and hence predictive computations during sentence comprehension.

I think that this is an informative and potentially important paper. It introduces new methods and findings into an area of ongoing theoretical interest concerning the role of predictive processes in sentence comprehension. As the authors are aware, however, such strong conclusions concerning predictive processes remain controversial. The reviewers of the paper were all in agreement that additional analyses were required to rule out some alternative factors that might also explain the observed findings.

Essential revisions:

1) The most critical point which I'd ask you to address is to rule out some alternative explanations of your observations. The manuscript already includes some additional analyses in which comparisons are made between "within-pair" vs "within-category/between-pair" sentences. However, the reviewers were concerned that this is only one of many alternative factors that could explain the observations.

The reviewers suggested, and I agreed, that there are a number of additional factors that could lead to similar activation on the penultimate word of the sentence (SFW-1) and which must be ruled out if you are to conclude that your effect is driven by prediction of the sentence final word (SFW). Among the factors suggested by reviewers, include:

– form-based properties of the penultimate word (word length, orthographic or phonological similarity, etc)

– lexical/semantic properties of the penultimate word (syntactic class, word frequency, imageability/concreteness, etc etc)

– incidental properties of the sentence up-to and including the penultimate word (e.g. number of words up to that point, syntactic complexity – e.g. number of clauses, embedded dependencies, etc)

To address this point you can do two things: (1) assess whether these factors are more similar for your within-sentence pairs than for your between-sentence pairs, (2) assess whether increased neural similarity would still be observed in your MEG data when these other factors are matched.

As you might be thinking, though, performing these second analyses will be difficult using the methods in the present manuscript. You face a severe loss of power/sensitivity as you progressively divide the materials into smaller and smaller subsets. (Incidentally, I felt that you don't need to match for the number of items compared, though I'm sure there are others who would be reassured by these analyses).

I'd therefore like to suggest an alternative method for running these analyses and one which doesn't require you to run analyses on subsets of materials. The method was (AFAIK) introduced in a paper by Carlin et al. (2011, Current Biology):

https://doi.org/10.1016/j.cub.2011.09.025

This is a method in which you partial out extraneous or unmatched factors while performing RSA analysis. In Carlin's case for face perception, this was achieved using partial spearman correlations to rule out physical features when comparing gaze direction. However, no doubt other statistical methods (e.g. multiple linear regression, etc) can also be applied. This comes under the rubric of "representational geometry analysis".

I think the more that you can do to ensure that other aspects of your sentence pairs, and the penultimate word of these sentence pairs, does not explain your observations the more satisfied readers of the paper will be that the only plausible explanation of your findings is that there is a neural signature of the predicted final word.

2) Even if these analyses confirm that other factors can't explain your findings, then you still need to explain what predictive pre-activation means giving that this neural effect is absent for the ~400ms immediately prior to the onset of the sentence final word. To my mind, this does not negate your conclusions regarding lexico-semantic prediction – but it does mean that a more nuanced mechanism must be involved. In particular, the idea that this reflects "activity silent working memory" or pre-activation of specific lexical-semantic properties of target words seemed like a stretch to one of the reviewers and I would agree. It might be, though, that by considering similarities between orthographic or semantic properties of the SFW in different sentences the authors could provide additional evidence in this regard. In the absence of this, though, I think that the authors should explain that their findings are consistent with pre-activativation while acknowledging the degree to which other interpretations (e.g. integration) might be possible.

3) One further point that two of the reviewers were confused by – and I think must be clarified – concerns the order of presentation and blocking of sentence presentation. Is it the case that the sentence pairs were presented on successive trials? I would hope not, since this is a serious confound for RSA analyses in fMRI and could also be problematic for MEG. I think it's the case that trials for the within-sentence pairs are no closer together in time than a randomly selected between-sentence pair, but I couldn't see this unambiguously stated in the manuscript. I'd like for you to confirm this and (if possible) report further analyses in which temporal distance or temporal order of trial pairs is excluded as a nuisance factor in neural similarity analyses.

These three points should be the main focus of a revision to the manuscript. In addition to these main points, I've also appended the three reviews that I've received. These include many other minor points, suggested changes and requests for clarification which you would do well to heed. There's always scope to clarify methodological aspects of a complex study like this.

However, one methodological concern (from reviewer 3), which I'll not insist on you addressing, concerns the separation of spatial and temporal RSA methods. I agree that spatio-temporal analyses in single-subject source space could detect neural effects that are missed by their sensor space and time-based analyses. However, to my mind this methodological concern could explain the absence of some effects in the existing analyses, but not the presence of reliable effects. Since there's a lot of work involved, and substantial correction for multiple comparisons would reduce sensitivity, I'm going to give you the option of not performing these analyses and instead discussing potential limitations and future directions that could be taken in similar future work.

*Reviewer #1*:

This manuscript by Wang et al. reports an MEG study in the field of neurocognition of language, investigating whether or not lexico-semantic properties of specific words are 'pre-activated' in a constraining sentence context. To this end, the authors used an analysis approach based on representational similarity analysis. Participants read sentences, presented one word at a time (stimulus presentation time 200 ms; SOA 1000 ms). Sentences were presented in pairs, and constructed such that both sentences of a pair constrained for the same sentence final word (e.g., baby). Analyses focused on brain signals -2000 to +1000ms relative to the onset of the sentence-final word. A spatial similarity time course was calculated for each pair, by calculating the correlation between a vector of activation values per channel, at each time point, between the two sentences of the pair ('within pair' condition). The same was repeated for each sentence relative to each other sentence, i.e., relative to all sentences not in the pair, yielding similarity time courses for pairs of sentences predicting different SFWs ('between pair' condition). The authors' expectation was that these similarity indices should be higher in the within condition, i.e., between the two sentences constraining for the same word, as compared to the 'between' condition. There are two results in this paper: The authors first observed an increased similarity independent of condition, roughly between 100 and 500 ms post onset of each word (i.e., for the sentence final word/SWF itself as well as the two preceding words, falling into the analysis time window, i.e., SWF-1 and SWF-2). Investigating further these time windows, the authors observed an increased similarity for within as compared to between pairs on the word preceding the critical word, i.e., SWF-1 (120 to 515 ms post onset SWF-1 or -880 to -485 ms relative to SWF; Figure 2B). By generalizing this analysis over time, i.e., by correlating each time point of the epoch with activity vectors from each other time point of the epoch, the authors show that this increased similarity was temporally restricted and did not persist into the 800 ms inter-trial interval preceding the sentence final word (Figure 2D). As control analyses, the authors demonstrate that the same results are obtained (albeit somewhat weaker), when matching the number of different pairs to the number of similar pairs. Also, the result could not be accounted for by noun vs. verb (i.e., word category level) similarity. Finally, the authors implemented a similar approach for temporal similarity, i.e., the similarity over time of activation time courses, and localized this effect to lateral and medial left temporal regions, extending into cerebellum. The authors conclude that they demonstrate the activation of unique spatial and temporal patterns of brain activity associated with the pre-activation of specific lexico-semantic items (i.e., words) during language processing, and that this pre-activation was transient and dynamic.

General Evaluation

This study approaches an important and also timely research question, as it is currently strongly debated whether and how predictive processes are involved in language processing. The application of RSA to this question is also innovative and, as far as I can tell, technically implemented in an excellent way. However, this manuscript leaves several questions open which I will detail below, and certain aspects of the study design, in my view, call into question the validity of interpreting the reported results as a predictive pre-activation of words.

Major Points

Most importantly, I think that the authors provide no evidence to actually support their claim that the observed increased similarity among within-pair sentences, at the pre-final word, actually reflects the pre-activation of the sentence final word. I think the most plausible account is that context-dependent constraint is already high on the pre-final word, so that alternative interpretations like ease of integration of the pre-final word are at least as likely as the pre-activation account that the authors try to propose here. At the very least, these two alternative accounts have to be discussed equally; if the authors give preference to the pre-activation account, this should be grounded in reliable additional empirical evidence. (In the Discussion section, the authors argue that 'greater similarity between sentence contexts' is an unplausible account for their result, given that the two sentences of each pair were composed of different words and, in particular, the SFW-1 prefinal word always differed. However, the sentences were constructed such that the constraint was high, so this does not rule out the possibility of expectation / ease of integration effects on the prefinal word, in my opinion.)

Related to this, it is inconsistent with their interpretation that the similarity effect disappears with the offset of the word-induced brain response of the pre-final SWF-1 word, i.e., around 500 ms prior to the onset of the sentence final target word. A true prediction / pre-activation should persist. The a-posteriori interpretation based on 'activity silent working memory' is a vast over-interpretation of the results, based on no data. Also the proposal that the pre-activation may involve a sequence of activation of different lexico-semantic properties of the target word is speculative beyond the interpretation of results, and not grounded in any data.

Also related to this, I think that the control analysis testing for noun/verb differences is not sufficient to warrant the general claim that 'higher order grammatical and semantic' effects? differences? cannot influence the present result. The noun/verb category difference is just one of many such features. I would find it much more convincing if the authors could show in their stimulus materials, that no such differences exist on the target position in the critical as compared to the control contrasts, as well as for the positions preceding the sentence final word. Also, I think behavioral testing could easily quantify the degree of expectancy/constraint on the pre-final words. This kind of additional data would allow for a more empirically grounded interpretation of the similarity effect on the pre-final word.

Also related to this, I wonder whether there should not be similarity effects also on the target words itself. Even though at the target word position not the same words were presented (e.g., baby and child in a sentence pair both constraining for baby), the similarity between those is still substantially higher than, e.g., baby and fridge (example from Figure 1).

Even more so, should not the pre-activation lead to increased similarity between the pre-word period and the brain activation elicited during the actual presentation of the word itself?

It is unclear to me, why sentences were presented in pairs. The pair-wise presentation of 120 sentence pairs each constraining for the same sentence-final word, without doubt can induce strategic expectation effects towards the end of the sentence – which has nothing to do with the kind of highly automatized predictive processes as postulated in the predictive coding framework. I think that this problem is not solved by the fact that only one of the two sentences contained the constrained-for target word at the SWF position. Actually, the fact that one pair contained an unexpected word could even increase the strategic handling of these sentence pairs.

It is also unclear to me why the very slow and un-naturalistic presentation rate was chosen. Again, I think that this can induce strategic processes, as well as increased working memory load, which may influence the RSA results. (I also tend to think that this design was chosen as the ITI preceding the SFW is the most obvious time window to search for predictive pre-activation, see my first point above. Given that no results were reported for this 'silent' pre-word time window, I tend to be very critical about interpreting the results as predictive pre-activation of the sentence final word.)

Combined, these points suggest to me that the authors interpret their results too strongly. Predictive pre-activation is claimed in the title, Abstract, and Discussion. I think the authors should generally tone down these claims and provide a more realistic and balanced account for their interesting result.

In their control analysis, the authors demonstrate that the within-pair similarity is also higher than the similarity calculated on the remaining sentences within target words with nouns or verbs. They use this to claim that their result cannot spuriously result from higher-order syntactic or semantic effects. I think this control analysis is nice, but its interpretation goes way too far, as the authors only tested one of many possible such linguistic features. Also, it is unclear why this post-hoc analysis is necessary at all, if (as I expect) authors controlled stringently for such obvious differences in their item construction. Even is the latter were not the case, it should be possible to a posteriori select the sentences for the between-pair analysis, out of all possible combinations, such that they are optimally matched to the 120 critical pairs?

Concerning the source analysis of the temporal similarity analysis: I am not expert enough in MEG beamforming to really judge this, but it appears to me that the source localization shown in Figure 3B does not seem to be a plausible generator of the scalp distribution of the difference effect shown in Figure 3A, left-most panel?

A lot of information about the stimulus construction and item materials is missing. The authors describe how sufficiently high cloze probability was assured in the sentences of the 120 pairs. However, many further aspects are important, like word category, word frequency, concreteness, etc. In particular, I think that it is important to assure that such obvious lexical and semantic properties are (a) balanced between the within-pair and the between-pair comparisons, as these are the final statistical contrast on which all interpretations are based; (b) that similar information is provided for the two words preceding the sentence-final word. (c) Furthermore, I think it is important to also provide data for the cloze probability of the pre-final words, in particular given that this is where the effect is found (see also my first point above).

Parts of the Discussion section and interpretation of the data are far too speculative, including the discussion of specific semantic properties that might be activated. For example, the authors write that "These findings provide strong evidence that unique spatial patterns of activity, corresponding to the pre-activation of specific lexical items, can be detected in the brain." I think this is not warranted given the presented data. I am picking out a few examples in the following:

The authors make several claims as to the specific nature of lexico-semantic preactivation, which are also not supported by the reported study: "… the particular spatial pattern of brain activity associated with the pre-activation of the word baby may have reflected the pre-activation of spatially distributed representations of semantic features such as little, cute, and chubby, while.… the pre-activation of the word roses may have reflected the pre-activation of semantic properties such as red and beautiful." This, in my view, is overly speculative and at the same time suggests to the superficial reader a level of detail that is by no means reached in this study.

Another example involves the claim that "this may be because different properties associated with particular words became available at different time. For example, the different semantic features (little, cute, chubby) associated with.… baby might have been recruited at different time points.", as a possible account why there were only effects along the diagonal. However, again, this is not grounded in any empirical data, and in tendency fails to acknowledge that also along the diagonal, there was no persistent effect beyond 500 ms pre-word onset.

With respect to the neural mechanism, the authors state that "the absence of an effect off the diagonal suggests that the spatial patterns associated with pre-activation evolved dynamically over time". However, there is no evidence to support this claim. In particularly when considering that there is also no persistent effect along the diagonal, it most likely indicates that there was no sustained pre-activation over time.

Reviewer #2:

This manuscript presents research aimed at investigating the hypothesis that specific words are pre-activated in the brain given a constraining semantic context. The authors test this hypothesis by presenting highly constraining sentences such that the final word in each sentence can be easily predicted. Moreover, they do so such that pairs of sentences are likely to be predicted to finish with the same word. They then examine the similarity of spatial and temporal patterns in MEG preceding the presentation of the final words. In particular they compare the similarity in these patterns between pairs of sentences with the same final predicted word and pairs of sentences with different final words. They find that both the spatial patterns and temporal patterns are MORE similar for sentences where the same final word is predicted than for sentences where different final words are predicted. They take this as evidence that specific lexico-semantic predictions are made by the brain during language comprehension.

This was a very well designed piece of research with interesting and compelling results. The manuscript was well written and the discussion seemed reasonable.

I have a few relatively minor comments and queries:

1) The nice study design included ensuring that the paired sentences didn't actually finish in the same word and that sometimes the sentence with an unexpected word would appear first and sometimes the sentence with the expected word would appear first. The authors argue that this means the results are not simply explainable on the basis that subjects might retain the expected final word in memory when reading the second sentence of a pair. However, it seems to me that, even though the unpredicted word has a much lower cloze, the subject might still retain that unexpected word in memory when hearing the second sentence of a pair. It doesn't seem that likely to me, but it's conceivable. I mean when a subject hears the unexpected word 'child', they might be more likely to retrieve that word when they are next presented with a sentence for which 'baby' is the "correct" prediction, but for which 'child' is a reasonable final word. So, much and all as I like the design, I do think it is still possible that retrieval of a previously stored word is still possible. One thing that I was unclear on (and sorry if I just missed it) was the actual ordering of the sentence presentation. Did the two members of a pair of sentences always appear consecutively? If so, this would make the idea of retrieval even more likely. If the 120 sentences are all just presented in a random order, then I guess it is unlikely. Again, sorry if I missed that.

2) A minor query – were there different numbers of words in the sentences? Or always the same? And, relatedly, did the subject always know when the final word was going to appear? It's just that a pet worry of mine is the generalizability of language research done on isolated sentences that are very regular in their makeup. I imagine subjects get into an unusual mindset with linguistic processes overlapping with more general decision making strategies that may confound things. I don't think that's an issue here for two reasons: 1) it wouldn't explain why the data are more similar within sentences than between and 2) subject didn't have to make deliberative decisions at the end of each sentence. But still, it would be nice to get a sense of the variability (or lack of it) in the structure of the sentences.

3) Very minor – in subsection “Design and development of stimuli” there seem to be 109 pairs of sentences above 70% close and 12 that were lower. That makes 121, not 120.

4) In subsection “MEG data processing” the authors say "Within this 4000ms epoch, trials contaminated…were…removed…" How is there a trial within this epoch? Is the trial not the entire epoch? Or am I misunderstanding what you mean by a trial?

Reviewer #3:

This study investigates the process of predictive access to upcoming strongly constrained words during reading. It does so by combining spatial and temporal resolution of the MEG with the representational similarity analysis (RSA) and asking whether the sentences that strongly constrain to semantically synonymous words show greater similarity of activity patterns (within-pairs) than those constraining to semantically unrelated words (between-pairs). Results point to the LH inferior and medial temporal areas showing greater activation similarity for within-pairs before the perceptual onset of the critical word and are therefore candidate locations for the word-related pre-activations. This approach constitutes a novel and exciting contribution to the study of predictive processing/coding in the language domain.

My two main concerns are as follows. First, that the specific method used for the RSA analysis -separating spatial and temporal dimensions of the data and using one dimension (spatial) to narrow down the testing time-window of the other (temporal) also narrowed down the scope of the effects uncovered. Second, that the sentences within- and between-pairs were insufficiently matched in terms of the syntactic and lexicosemantic characteristics of the words directly preceding the critical predicted word and observed effects could have been related to those differences rather than pre-activations. These two issues would need to be addressed before the strength and interpretation of the current set of effects could be fully evaluated.

Major points:

1) My main concert in terms of analysis methods is that the separation of the temporal and spatial components of the RSA analysis unnecessarily limited the kind of effects that were uncovered. For calculation of both spatial similarity time-series and cross-temporal spatial similarity matrixes all MEG sensors were included and hence the effects would be greatest in the time-points where many sensors simultaneously show similar activity for pairs of sentences. This means that strong and extended in time but spatially localised (or insufficiently distributed) effects might be missed. Especially since for determining the significant time-windows, vectors were averaged across subjects, which means that localised effects had even less chance of surviving given that the same effects in different subjects could appear in different sensors – due to the differences in head shape etc. Further the concern for the temporal RSA is that the time-window where the effects were tested (-880 -485, SFW aligned) were derived on the basis of the spatial analysis, where spatially continuous differences between within- and between-pair similarity values were found after applying an arbitrary cut-off (r>0.04).

To avoid these issues a spatiotemporal RSA could be carried out in the source space directly, or firstly in the sensor space (across sensors and time points) and then the significant spatiotemporal clusters could be source localised. For example, if beamforming is used to derive single-trial source estimates, then data RDMs can be derived using a modified version of the Searchlight approach (e.g Nili et al., 2014; Su et al., 2012 and 2014). For every trial, at every grid point and every time step (every 1/5/10 ms) a 3D data matrix is extracted consisting of activation from n of neighbouring grid points and n time-samples. Then for each pair of sentences predicting the same trials these data matrixes are correlated. Then the effects are averaged across all within-pairs producing grid point by time point spatiotemporal correlation values for the within-pair condition. The same can be repeated for between-pairs. Then a pair t-test can be done to compare within- and between- data across time and grid space, significant spatiotemporal clusters of differences would be determined with cluster permutation. If no major effects have been missed by separating spatial and temporal dimensions of the data, then spatiotemporal RSA would further validate the current set of results.

2) I have several questions about the experimental stimuli. Firstly, were the experimental sentences both between and within-pairs controlled for sentence length (n of words) and syntactic complexity (n of clauses, presence of embedded dependences)? The issue would arise if, for example, all within-pairs happened to have the same syntactic structure/complexity, while between-pairs had mismatching or different structure/complexity. Then the increases of the similarity before SFW for the within-pairs could potentially be attributed to similar demands of grammatical/syntactic processing, while decreased similarity for between-pairs would be driven by differences in these processing demands. The authors cover this potential caveat in subsection “Unique spatial patterns of neural activity are associated with the prediction of specific

words, prior to the appearance of new bottom-up input” of the discussion, and argued that in this case we would see within- and between-pair difference arise earlier. However, while such differences could have been building up, they also could have become significant only closer to the end of the sentence. To exclude this option differences between within- and between-pair sentences should be reported.

Secondly, were the SFW-1 words (the word directly before the SFW) controlled for any of the following characteristics across conditions: syntactic class, frequency, any semantic characteristics such as imageability, concreteness? Again, if the within-pairs matched in terms of SFW-1 characteristics more than the between-pairs sentences effects in the 'prediction' time-window could be driven by similarities of the SFW-1 processing and not the by the SFW pre-activation. Since the critical claim of this paper is that increases in spatial and temporal correlation of the neuronal activity for the averaged within-pairs is driven by pre-activation of the SFW it is critical to exclude any of the effects described above.

3) This point is related to the conclusions drawn by the authors in the Discussion section about the nature of the pre-activated representations. The authors suggest that the effects observed in the pre-SFW window can be driven by orthographic or phonological features of the predicted words. Have any of the analyses they proposed (subsection “Unique spatial patterns of neural activity are associated with the prediction of specific words, prior to the appearance of new bottom-up input”) been carried out? Since sentences used for this study were indeed very constraining, SFW pre-activation of the perceptual features of strongly predicted words would be expected under the predictive processing/coding approach.

[Editors' note: further revisions were requested prior to acceptance, as described below.]

Thank you for resubmitting your work entitled "Specific lexico-semantic predictions are associated with unique spatial and temporal patterns of neural activity" for further consideration at *eLife*. Your revised article has been favorably evaluated by Joshua Gold as Senior Editor and a Reviewing Editor. The reviewing editor writes:

I've now read the manuscript and author rebuttal in detail and I'm pleased to see that the authors have addressed the technical and methodological challenges that were raised by the reviewers of the original manuscript. I'm now satisfied that the results establish that information conveyed by a predictable sentence final word is activated at an early stage during processing of the penultimate word in a sentence. I think that this finding makes an important contribution to the literature by more-firmly establishing early lexical-semantic activation of predicted words as a key property of human sentence processing, and demonstrating a novel method by which the these time-limited lexical-semantic predictions can be shown in neural data.

While the manuscript has been much improved there are two remaining issues that I think need to be addressed before acceptance, as outlined below:

1) There are too many places in the Introduction and Discussion in which I think the authors aren't thinking critically enough about whether it is only their preferred "generative and predictive" view that could explain the present findings. My view is that many other accounts could also explain their findings. Specifically, any model which: (i) activates a cumulative semantic representation of sentence meaning, and (ii) emphasises processing speed and efficiency such that semantic representations that are strongly implied by the words read so far, but not yet directly expressed in words are activated – can also account for the current findings. There are many such models in the literature, but most notable (to my mind) is the "sentence gestalt" model from St John and McClelland, 1990, that has been recently updated by Rabovsky et al., 2018, and can predict the magnitude of EEG N400 responses in a wide range of sentence processing paradigms. To my knowledge this is not a model which is explicitly "generative and predictive" and yet I think it very likely that RSA analysis of the sentence gestalt representations generated by this model could simulate the results of the present study. While I don't think that the authors need to do the work to explore whether the model *can* simulate their findings, I do think that it is in their interests to offer a more balanced overview of the literature and to more precisely explain what sort of computational model is implied by their findings.

2) I had one other minor question about the method that they used in comparing cloze probabilities between and within item pairs which could be addressed by same time. This point is described in more detail in their rebuttal letter than in the manuscript. However, I think that this issue deserves a little more attention in the manuscript given the known importance of cloze probability in predicting the magnitude of EEG/MEG signals during sentence processing, and the. Specifically, in the rebuttal letter the authors report analyses of the difference between cloze probability for sentence pairs. However, if my understanding of this analysis is correct this analysis should be conducted not on the difference between cloze probabilities, but rather the absolute difference between cloze probabilities for within and between item pairs. I think that otherwise the average difference between cloze values would always be zero. I'd like the authors to report this analysis in the manuscript, including a description of the method used for conducting the analysis.

---

## [Author Response]

Essential revisions:1) The most critical point which I'd ask you to address is to rule out some alternative explanations of your observations. The manuscript already includes some additional analyses in which comparisons are made between "within-pair" vs "within-category/between-pair" sentences. However, the reviewers were concerned that this is only one of many alternative factors that could explain the observations.The reviewers suggested, and I agreed, that there are a number of additional factors that could lead to similar activation on the penultimate word of the sentence (SFW-1) and which must be ruled out if you are to conclude that your effect is driven by prediction of the sentence final word (SFW). Among the factors suggested by reviewers, include:– form-based properties of the penultimate word (word length, orthographic or phonological similarity, etc)– lexical/semantic properties of the penultimate word (syntactic class, word frequency, imageability/concreteness, etc etc)– incidental properties of the sentence up-to and including the penultimate word (e.g. number of words up to that point, syntactic complexity – e.g. number of clauses, embedded dependencies, etc)To address this point you can do two things: (1) assess whether these factors are more similar for your within-sentence pairs than for your between-sentence pairs, (2) assess whether increased neural similarity would still be observed in your MEG data when these other factors are matched.As you might be thinking, though, performing these second analyses will be difficult using the methods in the present manuscript. You face a severe loss of power/sensitivity as you progressively divide the materials into smaller and smaller subsets. (Incidentally, I felt that you don't need to match for the number of items compared, though I'm sure there are others who would be reassured by these analyses).

We fully agree that it is important to rule out the possibility that differences in processing of the penultimate word of the sentence (SFW-1) led to the pattern of results we observed. We have now carefully addressed this possibility and we think we can make a strong case that the lexical properties or the predictability of the penultimate word cannot account for our findings. We have made several major changes to the manuscript as described below:

A) In the Materials and methods, we now state that we measured: (1) the number of words, the number of clauses, and the syntactic complexity of the sentence context up until and including SFW-1; (2) various lexical properties of the SFW-1 (visual complexity, word frequency, syntactic class); and (3) the predictability (as operationalized by cloze probability) of the SFW-1. We showed that none of these factors differed systematically between pairs of contexts that predicted the same SFW (i.e. *within-pairs*) and pairs of contexts that predicted a different SFW (i.e. *between-pairs*).

We were unable to examine the orthographic or phonological features of the SFW-1 as a whole, because, in Chinese, the characters within each word/phrase are associated with distinct orthographic and phonological features. Also, as shown in the full set of stimuli (Figure 1A—source data 1), the SFW-1 could either be a content word (verb, noun, adjective, adverb) or a function word (pronoun, classifier, conjunction, particle, prepositional phrases). Concreteness values for these words were not available in available Chinese corpora. However, given the heterogeneity of the SFW-1, we think that the concreteness of the SFW-1 is unlikely to account for the observed effect.

B) In the Results, we describe a new control analysis that we carried out in order to fully exclude the possibility that the increased spatial similarity associated with sentence pairs that predicted the same SFW versus a different SFWwas driven by processing of the SFW-1 rather than anticipatory processing of the SFW itself. In this control analysis, we selected a subset of *between-pair* sentences that contained exactly the same SFW-1, but nonetheless predicted a different SFW. We then selected sentences that constrained for these same SFWs (*within-pairs)*, but which differed in the SFW-1. We then compared the spatial similarity between these two subsets of sentence pairs. If the increased spatial similarity associated with the *within-pairs* versus *between-pairs* was due to the lexical processing of the SFW-1, then the spatial similarity should be greater in sentence pairs containing exactly the same SFW-1 (i.e. in the subset of *between-pairs*) than in sentence pairs that predicted the same SFW (i.e. in the subset of *within-pairs*). We found no evidence for this. Instead, the spatial similarity remained larger for the *within-pairs* than the *between-pairs* (although in this subset analysis, the difference only approached significance due to limited statistical power).

C) In the Discussion, we now explicitly discuss all these methods (described above) to address the possibility that differences in processing of the penultimate word of the sentence (SFW-1) led to the pattern of results we observed.

I'd therefore like to suggest an alternative method for running these analyses and one which doesn't require you to run analyses on subsets of materials. The method was (AFAIK) introduced in a paper by Carlin et al. (2011, Current Biology):https://doi.org/10.1016/j.cub.2011.09.025This is a method in which you partial out extraneous or unmatched factors while performing RSA analysis. In Carlin's case for face perception, this was achieved using partial spearman correlations to rule out physical features when comparing gaze direction. However, no doubt other statistical methods (e.g. multiple linear regression, etc) can also be applied. This comes under the rubric of "representational geometry analysis".I think the more that you can do to ensure that other aspects of your sentence pairs, and the penultimate word of these sentence pairs, does not explain your observations the more satisfied readers of the paper will be that the only plausible explanation of your findings is that there is a neural signature of the predicted final word.

We thank the editor for the suggestion. We carefully read the paper recommended. However, these methods are not easily adapted for the way we chose to carry out our analysis. Specifically, in Carlin et al.’s 2011 study, the authors correlated *item* pairwise dissimilarity matrices. One matrix reflected the dissimilarity of the brain activity for all pairs of stimuli, and the other matrix reflected the dissimilarity of the factor of interest (i.e. qualitative gaze direction) for all pairs of stimuli. They also built dissimilarity matrices for other factors (such as grayscale intensities, head view, quantitative differences between angles of left and right gaze). This then allowed them to run partial Spearman correlations between the matrix reflecting the brain activity and the matrix reflecting the factor of interest, while controlling for the other factors on each item.

However, in the current study, we calculated the *means* across items of the brain pattern similarity values based on whether the same SFW was predicted by pairs of sentences (*within-pair*: the same SFW was predicted; *between-pair*: a different SFW was predicted). Our analysis approach has the advantage of increasing the signal-to-noise ratio of our correlation values, since the correlation values produced by random noise would be canceled out after averaging across items. Power is a particularly important consideration given that neural activity associated with the prediction of complex lexico-semantic representations (examined here) are likely to be smaller than activity associated with the perception of lower-level stimuli (probed in Carlin et al.’s study). However, our analysis approach makes it difficult to run correlation analyses that partial out other extraneous variables. Nevertheless, the additional control analysis described above, along with the additional measures of the contexts and the SFW-1, increase our confidence in claiming that the pattern of results we observed was driven by anticipatory processing of the SFW itself.

2) Even if these analyses confirm that other factors can't explain your findings, then you still need to explain what predictive pre-activation means giving that this neural effect is absent for the ~400ms immediately prior to the onset of the sentence final word. To my mind, this does not negate your conclusions regarding lexico-semantic prediction – but it does mean that a more nuanced mechanism must be involved. In particular, the idea that this reflects "activity silent working memory" or pre-activation of specific lexical-semantic properties of target words seemed like a stretch to one of the reviewers and I would agree. It might be, though, that by considering similarities between orthographic or semantic properties of the SFW in different sentences the authors could provide additional evidence in this regard. In the absence of this, though, I think that the authors should explain that their findings are consistent with pre-activativation while acknowledging the degree to which other interpretations (e.g. integration) might be possible.

We agree that the timing of the spatial similarity effect deserved more discussion. To our minds, there were two interesting features of this timing. The first is that the spatial similarity effect began to appear soon after the onset of SFW-1 (rather than at the offset of the SFW-1). The second is that the effect was not seen immediately prior to the appearance of the SFW itself. We consider each of these below:

A) The early onset of the spatial similarity effect.

The fact that the spatial similarity effect began to appear immediately following the SFW-1 raises the obvious question of whether the effect was driven by the lexical properties of the SFW-1 itself, or the predictability of the SFW-1 itself. For example, if the predictability of the SFW-1 differed systematically between pairs of contexts that predicted the same SFW (i.e. *within-pairs*) and pairs of contexts that predicted a different SFW (i.e. *between-pairs*), then the spatial similarity effect might have been driven by the “integration” of the SFW-1 rather than the prediction of the SFW. However, given that, as discussed above, the cloze values of the SFW-1 did not differ at all between pairs of contexts that predicted the same SFW (i.e. *within-pairs*) and pairs of contexts that predicted a different SFW (i.e. *between-pairs*), this seems unlikely. Moreover, the control analysis described above excludes the possibility that the spatial similarity effect was driven by bottom-up processing of the SFW-1 itself. This is discussed in subsection “Unique spatial patterns of neural activity are associated with the prediction of specific words, prior to the appearance of new bottom-up input”.

Rather than reflecting processing of the SFW-1 itself, we now make it clear in the Discussion that the early appearance of the effect “provides evidence that the prediction of the SFW was generated at the first point in time at which participants had sufficient information to unambiguously generate this prediction. For example, in the sentence “In the crib, there is a sleeping …”, as comprehenders accessed the meaning of the word, <sleeping>, they may have also predicted the semantic features of <baby>. This type of account follows from a generative framework of language comprehension in which, following highly constraining contexts, comprehenders are able to predict entire event or states, along with their associated semantic features, and incorporate such predictions into their mental models prior to the appearance of new bottom-up input (see Kuperberg and Jaeger, 2016; Kuperberg, 2016).”

B) The disappearance of the spatial similarity effect immediately prior to the appearance of the SFW itself.

In the Discussion, we now more explicitly state that the precise reason for this is unclear. First, we acknowledge that it is possible that the predicted information was not maintained over the relatively long interstimulus interval used in the present study. On the other hand, we also point out that a failure to detect neural activity over a delay does not necessarily imply that this information was not present. There is now quite compelling evidence from related fields challenging traditional ideas of how information is represented in the brain over delays. Having re-read the papers that we cited, we continue to think that they provide an interesting and possible explanation for why we did not see any effect in the delay period — one that we would like our readers to consider. We have tried to be more explicit about this, explaining that “representations within working memory can be maintained in a silent neural state, instead of being accompanied by persistent delayed activity (Stokes, 2015; Wolff et al., 2017). Such content-specific silent activity can only be detected if it is in the focus of attention and task-relevant. On this account, in the present study, despite the fact that we were not able to detect it, the predicted information was still present during the interstimulus interval, and it only became available once new bottom-up input was encountered.” Moreover, we pointed out that “Of course, this interpretation is speculative, particularly given our use of a very slow presentation rate. It will be important for future work to determine whether similar dynamics are associated with the prediction of upcoming words when bottom-up inputs unfold at faster, more naturalistic rates.”

3) One further point that two of the reviewers were confused by – and I think must be clarified – concerns the order of presentation and blocking of sentence presentation. Is it the case that the sentence pairs were presented on successive trials? I would hope not, since this is a serious confound for RSA analyses in fMRI and could also be problematic for MEG. I think it's the case that trials for the within-sentence pairs are no closer together in time than a randomly selected between-sentence pair, but I couldn't see this unambiguously stated in the manuscript. I'd like for you to confirm this and (if possible) report further analyses in which temporal distance or temporal order of trial pairs is excluded as a nuisance factor in neural similarity analyses.

We apologize for the confusion.

A) We have now made this clearer in the Introduction, Results as well as in the Materials and methods. In the Introduction, we state that “During the experiment, sentences were presented in a pseudorandom order, with at least 30 other sentences (on average 88 sentences) in between each member of a given pair.” In the Results, we state that “The sentences were constructed in pairs (120 pairs) that strongly predicted the same sentence-final word (SFW), although, during presentation, members of the same pair were separated by at least 30 (on average 88) other sentences.” In the Materials and methods, we state that “the two members of each pair were presented apart from each other, with at least 30 (on average 88) sentences that predicted different words in between.”

B) As mentioned above, our analysis methods make it difficult to explicitly account for the temporal distance between members of *within-pair* sentences as a nuisance factor. However, as noted in the previous version of the manuscript, we did carry out a control analysis, which “found that the spatial similarity effect was just as large when the unexpected SFW of a pair was presented before the expected SFW, as when the expected SFW was presented first (see Figure 2—figure supplement 3).”

C) We also state in the Discussion that “It is, however, conceivable that participants recognized a match between the word that they had just predicted and a word that they had predicted earlier in the experiment (even though this predicted word was never observed). For example, there is some evidence that a predicted SFW can linger in memory across four subsequent sentences, even if it is not actually presented (Rommers and Federmeier, 2018). This seems less likely to have occurred in the present study, however, where each member of a sentence pair was separated by at least 30 (on average 88) other sentences.”

These three points should be the main focus of a revision to the manuscript. In addition to these main points, I've also appended the three reviews that I've received. These include many other minor points, suggested changes and requests for clarification which you would do well to heed. There's always scope to clarify methodological aspects of a complex study like this.

We appreciate your careful summary of the reviewers’ concerns, and we hope that we have addressed them clearly.

However, one methodological concern (from reviewer 3), which I'll not insist on you addressing, concerns the separation of spatial and temporal RSA methods. I agree that spatio-temporal analyses in single-subject source space could detect neural effects that are missed by their sensor space and time-based analyses. However, to my mind this methodological concern could explain the absence of some effects in the existing analyses, but not the presence of reliable effects. Since there's a lot of work involved, and substantial correction for multiple comparisons would reduce sensitivity, I'm going to give you the option of not performing these analyses and instead discussing potential limitations and future directions that could be taken in similar future work.

We thank the editor for the suggestion. In the Results, we now point out that the analysis approach we took is fairly conservative. Specifically, we explain that “it was limited to the time window that showed a spatial similarity effect, and so it may not have captured more extended temporal similarity effects that were not accompanied by a spatial similarity effect.” We also point out that “The reason we took this approach is that we were interested, *a priori,* in any functional relationship between these measures, i.e. whether the spatial similarity effect reflected brain activity associated with the prediction of spatially distributed semantic representations, and whether the temporal similarity effect reflected brain activity associated with temporal binding of these spatially distributed representations. However, in order to fully exploit the spatiotemporal pattern of the data, future studies could examine the spatial and temporal patterns simultaneously using a spatiotemporal searchlight approach (Nili et al., 2014; Su et al., 2012; Su et al., 2014).”

Reviewer #1:Major PointsMost importantly, I think that the authors provide no evidence to actually support their claim that the observed increased similarity among within-pair sentences, at the pre-final word, actually reflects the pre-activation of the sentence final word. I think the most plausible account is that context-dependent constraint is already high on the pre-final word, so that alternative interpretations like ease of integration of the pre-final word are at least as likely as the pre-activation account that the authors try to propose here. At the very least, these two alternative accounts have to be discussed equally; if the authors give preference to the pre-activation account, this should be grounded in reliable additional empirical evidence. (In the Discussion section, the authors argue that 'greater similarity between sentence contexts' is an unplausible account for their result, given that the two sentences of each pair were composed of different words and, in particular, the SFW-1 prefinal word always differed. However, the sentences were constructed such that the constraint was high, so this does not rule out the possibility of expectation / ease of integration effects on the prefinal word, in my opinion.)

We thank the reviewer for encouraging us to consider these potential confounds more carefully. We have carried out several additional analyses and have made several changes to the manuscript to address the concern that differences in processing of the pre-final word (SFW-1) led to the pattern of results we observed.

A) Perhaps most relevant to the reviewer’s point that the results could be driven by “the possibility of expectation/ease of integration on the pre-final word”, we ran a new cloze test to examine the probability of the SFW-1 (now described in Materials and methods). We found that the cloze probability of the SFW-1 was relatively low: 11% on average across all items. Moreover, the difference in cloze probability between members of sentence pairs was matched between pairs that constrained for the same SFW *(within-pairs*: 17.00% cloze difference) and pairs that constrained for a different SFW (*between-pairs*: 17.28% cloze difference): t_(28678)_ = -0.136, p = 0.89. We think that this makes it unlikely that the observed effect was driven by the expectation or ease of integration of the SFW-1.

B) In the Materials and methods, we now state that we extracted: (1) the number of words, number of clauses, syntactic complexity of the sentence context up until and including SFW-1; (2) various lexical properties of the SFW-1 itself (visual complexity, word frequency, syntactic class). We show that none of these factors differed between pairs of contexts that predicted the same SFW (i.e. *within-pairs*) and pairs of contexts that predicted a different SFW (i.e. *between-pair*s).

C) In the Results, we now describe a new control analysis that we carried out in order to fully exclude the possibility that the spatial similarity effect was driven by lexical processing of the SFW-1, rather than anticipatory activity related to the prediction of the SFW itself. In this control analysis, we selected a subset of *between-pair* sentences (i.e. that predicted a different SFW) but that contained exactly the same SFW-1. We then selected a subset of *within-pair* sentences that constrained for these same SFWs, but that differed in the SFW-1. We then compared the spatial similarity between these two subsets of sentence pairs. If, in our original analysis, the increased spatial similarity associated with the *within-pair* sentences relative to the *between-pair* sentences was in fact driven by lexical processing of the SFW-1, then the spatial similarity should be greater in sentence pairs containing exactly the same SFW-1 (i.e. in the subset of the *between-pair* sentences) than in sentence pairs that predicted the same SFW (i.e. in the subset of *within-pair* sentences). We found no evidence for this. Instead, the spatial similarity remained greater in the sentence pairs that predicted the same SFW than in sentence pairs that predicted a different SFW (although in this subset analysis, the difference only approached significance due to the limited statistical power). We further discussed this finding in the Discussion in subsection “Unique spatial patterns of neural activity are associated with the prediction of specific words, prior to the appearance of new bottom-up input”.

Related to this, it is inconsistent with their interpretation that the similarity effect disappears with the offset of the word-induced brain response of the pre-final SWF-1 word, i.e., around 500 ms prior to the onset of the sentence final target word. A true prediction / pre-activation should persist. The a-posteriori interpretation based on 'activity silent working memory' is a vast over-interpretation of the results, based on no data.

We agree that the timing of the spatial similarity effect deserved more discussion. As the reviewer points out, there are two interesting features of this timing. The first is that the spatial similarity effect began to appear soon after the onset of the SFW-1 (rather than at the offset of the SFW-1). The second is that the effect then disappeared and was not detected immediately prior to the appearance of the SFW itself. We consider each of these points below:

A) The early onset of the spatial similarity effect.

The fact that the spatial similarity effect began to appear soon after the onset of the SFW-1 raises the obvious question of whether the effect was driven by either the lexical properties of the SFW-1 itself, or the predictability of the SFW-1 itself. As discussed above, we found no evidence that this was the case. Please see Results, Discussion and Materials and methods for details.

Rather, in the Discussion, we suggest that the early appearance of the effect “provides evidence that the prediction of the SFW was generated at the first point in time at which participants had sufficient information to unambiguously generate this prediction. For example, in the sentence “In the crib, there is a sleeping …”, as comprehenders accessed the meaning of the word, <sleeping>, they may have also predicted the semantic features of <baby>. This type of account follows from a generative framework of language comprehension in which, following highly constraining contexts, comprehenders are able to predict entire event or states, along with their associated semantic features, and incorporate such predictions into their mental models prior to the appearance of new bottom-up input (see Kuperberg and Jaeger, 2016; Kuperberg, 2016).”

B) The disappearance of the spatial similarity effect immediately prior to the appearance of the SFW itself.

In the Discussion, we now state more explicitly that the precise reason for this is unclear. First, we acknowledge that it is possible that, the predicted information was not maintained over the relatively long interstimulus interval used in the present study. On the other hand, we also point out that a failure to detect neural activity over a delay does not necessarily imply that this information is not present. There is now quite compelling evidence from related fields challenging traditional ideas of how information is represented in the brain over time delays. Having re-read the papers that we cited, we continue to think that they provide a possible explanation for why we didn’t see any effect in the delay period — one that we’d like our readers to consider. We have tried to be more explicit about this, explaining that the contents of working memory can be maintained in a silent neural state, instead of being accompanied by persistent delayed activity (Stokes, 2015; Wolff et al., 2017). We also now discuss the idea that content-specific silent activity can only be detected if it is in the focus of attention and task-relevant. On this account, in the present study, despite the fact that we were not able to detect it, the predicted information was still present during the interstimulus interval, and it only became available once new bottom-up input was encountered. Of course, this interpretation is speculative, particularly given our use of a very slow presentation rate. Therefore, we have emphasized that “It will be important for future work to determine whether similar dynamics are associated with the prediction of upcoming words when bottom-up inputs unfold at faster, more naturalistic rates.”

C) Finally, it is possible that, in the earlier version of the manuscript, there was some ambiguity in our use of the term “pre-activation” (of a lexico-semantic representation).

Some people have used the term “pre-activation” to refer specifically to the pre-activation of specific phonological or orthographic word-forms. We did not make this assumption. Rather, as we make clear in the Discussion, our assumption is that multiple different types and grains of information can be encoded (and therefore predicted) within a predicted lexical representation, and that it is unclear exactly what information was detected by our analysis.

To avoid any such ambiguity, in the revised version of the manuscript we now use the more general term “prediction” throughout. We explain what we mean by this at the very beginning of the Introduction: “After reading or hearing the sentence context, “In the crib there is a sleeping …”, we are easily able to predict the next word, “baby”. In other words, we are able to access a unique lexico-semantic representation of <baby> that is different from the lexico-semantic representation of any other word (e.g. <rose>), ahead of this information becoming available from the bottom-up input.” We also rephrased the introduction of the hierarchical generative framework of language comprehension: “strong beliefs about the underlying message that is being communicated can lead to the prediction of associated semantic features and sometimes to the top-down pre-activation of information at lower levels of the linguistic hierarchy (e.g. orthographic and/or phonological form) before new bottom-up information becomes available.”

In the Discussion, we state that “It is also possible that the increased spatial similarity in association with sentence pairs that predicted the same word reflected similarities of predictions generated at a lower phonological and/or orthographic level of representation. On this account, comprehenders not only predicted the semantic features of words, but they also pre-activated their word-forms. The present study cannot directly speak to this hypothesis.”

Later, we further re-iterate this point by stating that, while our findings provide evidence that the prediction of the semantic features of SFWs was generated at the first point in time at which participants had sufficient information to unambiguously generate this prediction, “we cannot tell from the current findings whether this, in turn, led to the top-down pre-activation of specific phonological or orthographic word-forms.”

Also the proposal that the pre-activation may involve a sequence of activation of different lexico-semantic properties of the target word is speculative beyond the interpretation of results, and not grounded in any data.

We found that the increased spatial similarity in the *within-pair* sentences, relative to the *between-pair* sentences, was only evident along the diagonal line; it did not generalize across time points. This suggests that the unique spatial pattern of brain activity associated with the prediction of specific words changed over time. In order to interpret this result, we speculated that “this may be because different properties associated with particular words became available at different times. For example, the different semantic features (e.g., <human>, <small>, <cries>) associated with the prediction of a specific word (e.g. “baby”) might have been recruited at different time points.”

Also related to this, I think that the control analysis testing for noun/verb differences is not sufficient to warrant the general claim that 'higher order grammatical and semantic' effects? differences? cannot influence the present result. The noun/verb category difference is just one of many such features. I would find it much more convincing if the authors could show in their stimulus materials, that no such differences exist on the target position in the critical as compared to the control contrasts, as well as for the positions preceding the sentence final word. Also, I think behavioral testing could easily quantify the degree of expectancy/constraint on the pre-final words. This kind of additional data would allow for a more empirically grounded interpretation of the similarity effect on the pre-final word.

We apologize for any confusion about this analysis.

A) This was not intended to be a “control” analysis. Instead, its purpose was to help us determine the type and grain of predicted information that might have been reflected by the item-specific unique spatial patterns. In our study, 50% of the predicted SFWs were nouns and 50% were verbs. Thus, in theory, an increased spatial similarity in association with sentence pairs that predicted the same upcoming word could have reflected greater similarity between the predicted word’s general syntactic category (a noun or a verb).

This analysis aimed to exclude this possibility. It therefore did not test for any *difference* in the prediction of nouns versus verbs. Rather, we averaged the spatial similarity values of sentence pairs that predicted nouns and verbs together, and we extracted the spatial similarity values of sentence pairs that predicted the same syntactic category (whether this was a noun or a verb), i.e. *within-category* sentence pairs. We then compared these *within-category* spatial similarity values with the original item-specific *within-pair* spatial similarity values. We found that the *within-category* spatial similarity values were significantly smaller than the *within-pair* spatial similarity values. These findings suggest that “the greater *within-pair* (versus *between-pair*) spatial similarity effect was not simply reducible to the prediction of general syntactic category”. This has been better explained in the Results section.

B) We ruled out the possibility that the item-specific prediction effect was driven by the differences in the pre-final words (SFW-1) or the preceding contexts, as we also discussed above. Detailed revisions were made in the Results, Discussion and Materials and methods.

C) We thank the reviewer for suggesting that we quantify the degree of expectancy/constraint on the pre-final words (SFW-1s). As noted above, we now report the results of a cloze probability test that examined the predictability of the SFW-1 (in the Materials and methods). We found that the cloze probability of the SFW-1 was relatively low: 11% on average across all items. Also, the difference in cloze probability of the SFW-1 was matched between the *within-pair* sentences (17.00% cloze difference) and the *between-pair* sentences (17.28% cloze difference): t_(28678)_ = -0.14, p = 0.89. We believe that this provides strong evidence that the observed effect was driven by the prediction of the SFW instead of the expectation or ease of integration of the SFW-1.

Also related to this, I wonder whether there should not be similarity effects also on the target words itself. Even though at the target word position not the same words were presented (e.g., baby and child in a sentence pair both constraining for baby), the similarity between those is still substantially higher than, e.g., baby and fridge (example from Figure 1).

We understand the reviewer’s point. However, there is a potential confound: it is well established that words that *violate* strong lexico-semantic predictions produce a larger amplitude response between 300-500ms (a larger N400), even when they are semantically related to predicted words (e.g. Federmeier et al., 1999). This was true in the present MEG data where we saw clear evidence of an increased N400 amplitude on unexpected SFWs over the left temporal sensors. An engagement of left temporal regions in processing unexpected SFWs in both members of a pair (regardless of whether these words are the same or different from each other) would inflate the estimate of the spatial similarity value on these words. This would confound the comparison of the spatial similarity values between *within-pair* versus *between-pair* SFWs because in 25% of the *between-pair* sentences, the SFW of both members of the pair was unexpected (i.e. (N*(N-1)/2 pairs out of 2*N*(N-1) pairs) whereas this was not true in any of the *within-pair* sentences. This would have inflated our estimate of the spatial similarity values of the *between-pair* sentences, thereby reducing our power to detect a significant difference between the *within-pair* and the *between-pair* SFWs.

Nonetheless, because examination of Figure 2B suggests that, after the onset of the SFW, the spatial similarity values indeed appeared to be slightly greater in sentence pairs that predicted the same SFW (*within-pairs*) than in sentence pairs that predicted a different SFW (*between-pairs*), we went ahead and compared the averaged spatial similarity between the *within-pair* and *between-pair* sentences within the time window of 109 – 588ms (defined by our cutoff threshold: R > 0.04) after the onset of the SFW. However, the difference was not significant: t_(25)_ = 1.388, p = 0.177. Given the confound and the complexity of any interpretation, we decided to focus the manuscript itself on activity prior to the onset of the SFW.

Even more so, should not the pre-activation lead to increased similarity between the pre-word period and the brain activation elicited during the actual presentation of the word itself?

This is an interesting question and one that we considered. However, addressing it runs into the same issues as those described above: any brain activity measured following the actual presentation of the SFWs is likely to reflect *both* information corresponding to the semantic features associated with the item-specific SFW itself, as well as more general processing of the SFW (regardless of its precise identity) in relation to its preceding context — that is, a SFW that is not predicted will evoke a larger N400 than a SFW that is predicted. This makes it tricky to interpret any similarities between brain activity produced prior the SFW and brain activity produced during the actual presentation of SFW, especially for unpredicted SFWs.

Despite this caveat, we carried out an exploratory analysis to examine the relationship between spatial patterns of activity produced during the prediction period and spatial patterns of activity produced following the onset of the SFW itself. We constructed two cross-temporal similarity matrices — one for expected SFWs (Author response image 1: left) and one for unexpected SFWs (Author response image 1: middle) — by correlating the spatial pattern of brain activity produced at each time point during the prediction window (-1000 to 0ms) with the spatial pattern of brain activity produced at each time point after the onset of the SFWs (0 to 1000ms). In order to determine whether there were any differences in the spatial pattern produced by unexpected SFWs and expected SFWs, we subtracted these two matrices (Author response image 1: right) and carried out a cluster-based permutation analysis. This revealed two effects (the two clusters shown in Author response image 1: right).

The first effect, shown in blue (Author response image 1: right), was driven by a stronger “pre-post” correlation in the sentences ending with *expected* than *unexpected* SFWs (cluster-level p < 0.001, 10000 permutations). Specifically, the spatial pattern of predictive activity that began just before the onset of the SFW, continuing until 200ms after its onset (-300 to 200ms) correlated with the spatial pattern of activity produced between 300-800ms *after* the onset of *expected* SFW. We speculate that, at -300ms prior to the onset of the SFW, the semantic features that had been predicted earlier were further activated in anticipation of the SFW, remaining active for 200ms after its onset; then, as the semantic features of the bottom-up expected input became available at around 400ms, they matched these predicted features thereby driving the increased spatial similarity values to the expected SFWs.

The second effect, shown in red (Author response image 1: right), was driven by a stronger “pre-post” correlation in the sentences ending with *unexpected* SFWs (cluster-level p = 0.025, 10000 permutations). Specifically, the spatial pattern of predictive activity that was originally produced following the SFW-1 (around 400-700ms prior to the onset of the SFW) correlated with the spatial pattern of brain activity produced between 400-800ms following the onset of *unexpected* SFWs. We speculate that the detection of the lexical violation on the SFW (during the N400 time window) triggered a re-activation of the originally predicted words, leading to the increased spatial similarity values within this later time window.

While these findings are interesting, these interpretations are obviously very speculative. Therefore, in the manuscript itself, we decided to focus on the well-motivated prediction effect preceding the SFW. However, we welcome the opportunity to share these preliminary data and our speculation here in this response to the reviewer’s question.

It is unclear to me, why sentences were presented in pairs. The pair-wise presentation of 120 sentence pairs each constraining for the same sentence-final word, without doubt can induce strategic expectation effects towards the end of the sentence – which has nothing to do with the kind of highly automatized predictive processes as postulated in the predictive coding framework. I think that this problem is not solved by the fact that only one of the two sentences contained the constrained-for target word at the SWF position. Actually, the fact that one pair contained an unexpected word could even increase the strategic handling of these sentence pairs.

A) We constructed these sentences in pairs (120 pairs) such that each member of a pair predicted the same word, even though their contexts differed (e.g. “In the crib, there is a sleeping …” and “In the hospital, there is a newborn …”). This was essential to the logic of our design — that the spatial similarity in brain activity produced prior to the onset of the SFW would be greater between members of a pair that predicted the same SFW than between members of a pair that predicted a different SFW. However, as we now emphasize in the Introduction, Results and Materials and methods, during the experiment itself, sentences were presented in a pseudorandom order, with at least 30 other sentences (on average 88 sentences) in between each member of a given pair.

B) Of course, even with this gap between the presentation of members of the same pair, it was important to avoid repetition confounds. This is why, during presentation of the stimuli, we replaced the SFW in one member of the sentence pair with an unpredicted word (therefore avoiding repetition of the SFW).

C) We also considered the possibility that participants may have been more likely to predict a particular word having previously seen this word. This is why we carried out the control analysis (subsection “Spatial RSA: The spatial pattern of neural activity was more similar in sentence pairs that predicted the same versus different words, and this effect began before the onset of the predicted word”), which showed that the spatial similarity effect was just as large when the unexpected SFW of a pair was presented before the expected SFW was presented as when they were presented in the opposite order (see Figure 2—figure supplement 3). In the Discussion (subsection “Unique spatial patterns of neural activity are associated with the prediction of specific words, prior to the appearance of new bottom-up input”), we now discuss in detail how our design might have affected the interpretation of the results.

D) We think that it is unlikely that the inclusion of these *unexpected* SFWs actually *increased* the strategic prediction effects towards the end of any given sentence. Previous studies have actually shown *reduced* effects of prediction when the validity of a predictive cue is low (e.g. during semantic priming: Lau, et al., 2013; Delaney-Busch et al., 2017; during sentence comprehension: Brothers et al., 2016; Brothers et al., 2017). In other words, this would have hurt our ability to detect an effect in the present study.

E) Having made these specific design points, we want to emphasize that we agree with the reviewer that these were far from naturalistic experimental conditions. As noted above, we see the unique contribution of our study as “providing evidence that, when we know that item-specific lexico-semantic are generated, they are associated with unique spatial and temporal patterns of neural activity”

In the Introduction, we are now more careful to emphasize up-front that our aim was to use “MEG, together with both spatial and temporal RSA, to ask whether, *under experimental conditions known to encourage specific lexico-semantic prediction*, distinct words are associated with distinct spatial and temporal patterns of neural activity, prior to the appearance of the predicted input.” And in the Discussion, we state that “these findings pave the way towards the use of these methods to determine whether and when such specific lexico-semantic representations become available as language, in both visual and auditory domains, unfolds more rapidly in real time.”

References:

Lau, E. F., Holcomb, P. J. and Kuperberg, G. R. (2013). Dissociating N400 Effects of Prediction from Association in Single-word Contexts. *Journal of Cognitive Neuroscience, 25*(3), 484-502

Delaney-Busch, N., Morgan, E., Lau, E., and Kuperberg, G. R. (2017). Comprehenders Rationally Adapt Semantic Predictions to the Statistics of the Local Environment: a Bayesian Model of Trial-by-Trial N400 Amplitudes. *CogSci*.

Brothers, T., Dave, S., Hoversten, L. J.,Traxler, M., Swaab, T. Y. (2016). Expect the unexpected: Speaker reliability shapes online lexical anticipation. Poster presented at the *8^th^ Society of Neurobiology of Language Conference*, London, England.

Brothers, T., Swaab, T. Y., and Traxler, M. J. (2017). Goals and strategies influence lexical prediction during sentence comprehension. *Journal of Memory and Language, 93*, 203-216. doi:https://doi.org/10.1016/j.jml.2016.10.002

It is also unclear to me why the very slow and un-naturalistic presentation rate was chosen. Again, I think that this can induce strategic processes, as well as increased working memory load, which may influence the RSA results. (I also tend to think that this design was chosen as the ITI preceding the SFW is the most obvious time window to search for predictive pre-activation, see my first point above.

Again, we agree with the reviewer that the experimental conditions in this study were un-natural. We have been more explicit in the Introduction that “the sentences were visually presented at a slow rate of 1000ms per word. This ensured the generation of specific lexico-semantic predictions and guaranteed sufficient time to detect any representationally specific neural activity before the onset of the predicted word.”

We return to this in the Discussion and have specifically pointed out that “It will be important for future work to determine whether similar dynamics are associated with the prediction of upcoming words when bottom-up inputs unfold at faster, more naturalistic rates.”

Given that no results were reported for this 'silent' pre-word time window, I tend to be very critical about interpreting the results as predictive pre-activation of the sentence final word.)

Please see above for our discussion of the timing of the observed effect. Note that we have made it clearer in the Discussion that “the predicted information was not maintained over the relatively long interstimulus interval used in the present study (SOA: 1000ms per word).”

Combined, these points suggest to me that the authors interpret their results too strongly. Predictive pre-activation is claimed in the title, Abstract, and Discussion. I think the authors should generally tone down these claims and provide a more realistic and balanced account for their interesting result.

As noted above, we no longer use the term, “pre-activation”, as it is possible that some people may interpret this as reflecting the pre-activation of a specific phonological or orthographic lexical form of a word. Instead, we use more general term, “prediction”, throughout the revised manuscript. Based on the design and all the control analyses we carried out, we think that our findings provide strong evidence that the prediction of semantic features associated with individual words produced unique spatial patterns of brain activity that were evident before new bottom-up input (i.e. the SFW itself) became available.

In their control analysis, the authors demonstrate that the within-pair similarity is also higher than the similarity calculated on the remaining sentences within target words with nouns or verbs. They use this to claim that their result cannot spuriously result from higher-order syntactic or semantic effects. I think this control analysis is nice, but its interpretation goes way too far, as the authors only tested one of many possible such linguistic features. Also, it is unclear why this post-hoc analysis is necessary at all, if (as I expect) authors controlled stringently for such obvious differences in their item construction. Even is the latter were not the case, it should be possible to a posteriori select the sentences for the between-pair analysis, out of all possible combinations, such that they are optimally matched to the 120 critical pairs?

In the Discussion, we have laid out several possible interpretations of the greater similarity associated with the *within-pair* versus *between-pair* sentences, ranging from the prediction of syntactic category, semantic features, to lower-level word-form features.

A) As discussed above, instead of controlling for the general syntactic category of the predicted SFWs, we explicitly manipulated this factor (i.e. the predicted SFWs could be verbs or nouns) so that we could ask whether the observed *within-pair* effect reflected *only* the prediction of syntactic category. We calculated the *within-category* spatial similarity between all pairs of sentences that predicted the same category of SFW. We compared these *within-category* spatial similarity values with the *within-pair* spatial similarity values. We found that the *within-category* spatial similarity values were significantly smaller than the *within-pair* spatial similarity values. These findings suggest that the *within-pair* versus *between-pair* spatial similarity effect did not simply reflect the prediction of these words’ broad syntactic category, but instead reflected prediction at the level of the semantic properties and features that defined the meanings of the predicted words, and perhaps lower-level orthographic and/or phonological features. This has been stated in the Discussion.

B) In the Discussion, we also further explain why the current study cannot address the question of whether or not the observed effects reflect the prediction of just the semantic features, or also the orthographic or phonological properties of the predicted words: “This is because, for the most part, there is a one-to-one correspondence between the semantic features and the phonological or orthographic forms of words. However, the methods described here provide one way of addressing this question in future studies. For example, by examining spatial similarity of sentence pairs that constrain for words with shared orthographic features but differing in their meanings (such as homonyms), it should be possible to dissociate the prediction of orthographic/phonological representations from the prediction of semantic features associated with a given lexico-semantic item.”

Concerning the source analysis of the temporal similarity analysis: I am not expert enough in MEG beamforming to really judge this, but it appears to me that the source localization shown in Figure 3B does not seem to be a plausible generator of the scalp distribution of the difference effect shown in Figure 3A, left-most panel?

The source localization in Figure 3B shows the *difference* between the temporal similarity associated the *within-pair* and the *between-pair* sentences. The scalp distribution shown in the left-most panel in Figure 3A indicates only the distribution of the temporal similarity values for the *within-pair* sentences. Therefore, the source localization (left inferior and medial temporal, extending to the cerebellum) in Figure 3B should be compared with the scalp distribution shown in Figure 3A right-most panel (most prominent over the central and posterior regions). In the revised Results, we have made this explicit: “The source localization of the difference (corresponding to the difference of the topographic distribution, see Figure 3A: right panel) is shown in Figure 3B.”

A lot of information about the stimulus construction and item materials is missing. The authors describe how sufficiently high cloze probability was assured in the sentences of the 120 pairs. However, many further aspects are important, like word category, word frequency, concreteness, etc. In particular, I think that it is important to assure that such obvious lexical and semantic properties are (a) balanced between the within-pair and the between-pair comparisons, as these are the final statistical contrast on which all interpretations are based; (b) that similar information is provided for the two words preceding the sentence-final word. (c) Furthermore, I think it is important to also provide data for the cloze probability of the pre-final words, in particular given that this is where the effect is found (see also my first point above).

A) We do not report the lexical features of the predicted SFW itself because any systematic difference in the lexical features of the *within-pair* predicted SFW (mean difference = 0) and the *between-pair* sentences (the mean difference will depend on the variability across items) is an intrinsic feature of our design.

B) As mentioned above, we extracted various lexical properties of the SFW-1 itself in all 240 of our sentences (visual complexity, word frequency, syntactic class). None of these factors differed systematically between pairs of contexts that predicted the same SFWs (i.e. *within-pairs*) and pairs of contexts that predicted different SFWs (i.e. *between-pairs*). We were unable to examine the concreteness or imageability of the SFW-1. This is because, as shown in the full set of stimuli (Figure 1A— source data 1), the SFW-1 could either be a content word (verb, noun, adjective, adverb) or a function word (pronoun, classifier, conjunction, particle, prepositional phrases). Concreteness values for these words were not available in available Chinese corpora. However, given the heterogeneity of the SFW-1, we think that the concreteness of the SFW-1 is unlikely to have had any effect on the observed effect.

C) As explained above, we also carried out a control analysis with a subset of stimuli with exactly the same SFW-1. This analysis revealed no evidence of an increased spatial similarity effect associated with lexical processing of the SFW-1.

D) As also noted above, we ran a separate cloze norming study to examine the probability of the SFW-1. We found that the cloze probability of the SFW-1 was relatively low: 11% on average across all items. Also, the difference in cloze probability of the SFW-1 was matched between the *within-pair* sentences (17.00% cloze difference) and the *between-pair* sentences (17.28% cloze difference): t_(28678)_ = -0.14, p = 0.89. We hope this provides sufficient evidence to rule out the possibility that the observed effect was explained by the expectation or ease of integration of the SFW-1.

E) Given that (1) the cloze study above suggested that the contextual constraint only became strong *after* the presentation of the SFW-1, and (2) we did not actually see any evidence of a spatial similarity effect following the SFW-2, we did not extract the lexical characteristics of the SFW-2 in all our sentences. However, we did extract the number of words, the number of clauses and the syntactic complexity of the sentence contexts up until and including SFW-1. Again, none of these factors differed systematically between pairs of contexts that predicted the same SFWs (i.e. *within-pairs*) and pairs of contexts that predicted different SFWs (i.e. *between-pairs*).

Parts of the Discussion section and interpretation of the data are far too speculative, including the discussion of specific semantic properties that might be activated. For example, the authors write that "These findings provide strong evidence that unique spatial patterns of activity, corresponding to the pre-activation of specific lexical items, can be detected in the brain." I think this is not warranted given the presented data. I am picking out a few examples in the following:The authors make several claims as to the specific nature of lexico-semantic preactivation, which are also not supported by the reported study: "… the particular spatial pattern of brain activity associated with the pre-activation of the word baby may have reflected the pre-activation of spatially distributed representations of semantic features such as little, cute, and chubby, while.… the pre-activation of the word roses may have reflected the pre-activation of semantic properties such as red and beautiful." This, in my view, is overly speculative and at the same time suggests to the superficial reader a level of detail that is by no means reached in this study.

In the revised version of the manuscript, we have been more careful in our wording to explain what we think that we can and cannot infer from these data. We continue to think that the data provide compelling evidence that “unique spatial and temporal patterns of neural activity are associated with distinct lexico-semantic predictions”. As discussed above, we now provide additional data and analyses to support this interpretation and to rule out an interpretation that the similarity effects observed were driven either by the lexical features or the predictability of the SFW-1.

As in any cognitive neuroscience study, we interpret our findings in relation to the prior literature. The reason why we designed this study in the first place — and why we think that the question is interesting — is because there was *a priori* reason to believe that the particular sets of semantic features associated with different words — or different predicted words — are associated with distinct patterns of spatial activity. As noted in the Introduction: “the various semantic features and properties associated with words and concepts are represented in the brain across spatially distributed multimodal networks (Damasio, 1989; Price, 2000; Martin and Chao, 2001) … For example, the particular set of semantic features and properties associated with the concept, <baby> (e.g. <human>, <small>, <cries>), might be represented by a particular spatially distributed pattern of neural activity, whereas the semantic features and properties associated with the concept, <rose> (e.g. <plant>, <scalloped petals>, <fragrant smell>) might be represented by a different spatially distributed pattern of neural activity.” This idea has a long history in cognitive neuroscience, and, as we also note, there is interesting evidence that it may be possible to capture evidence of distributed representations using spatial RSA (e.g. Devereux et al., 2013).

In the revised Introduction, we hope that we have explained the logic of our design more clearly, pre-empting our interpretation in the Discussion: “If, following a constraining context (e.g. In the crib, there is a sleeping …”), the prediction of a unique lexico-semantic item (<baby>) is represented by a unique spatial pattern of brain activity, then this spatial pattern should be more similar following another context that predicts the same word, i.e. *within-pair* (e.g. “In the hospital, there is a newborn …”) than following another context that predicts a different word, i.e. *between-pair* (e.g. On Valentine’s day, he sent his girlfriend a bouquet of red …”)”.

In the Discussion itself, we are more careful to make it clear that this is an *interpretation* of our results: “Instead, we *suggest that* the spatial similarity effect reflected similarities at the level of the semantic properties and features that defined the meanings of the predicted words … We *suggest* that our analysis picked up distinct spatially distributed patterns of neural activity that corresponded to the particular sets of features associated with distinct predicted words. For example, the prediction of the set of semantic properties and features corresponding to the word <baby> (e.g. <human>, <small>, <cries>) may have been reflected by the activation of a particular spatially distributed network that differed from the network reflecting the prediction of the set of semantic features corresponding to a different predicted word, <roses> (e.g. <plant>, <scalloped petals>, <fragrant smell>).”

We also offer other interpretations, e.g. that the patterns may have reflected the pre-activation of unique representations of the orthographic or phonological form of specific predicted words (Discussion section).

Another example involves the claim that "this may be because different properties associated with particular words became available at different time. For example, the different semantic features (little, cute, chubby) associated with.… baby might have been recruited at different time points.", as a possible account why there were only effects along the diagonal. However, again, this is not grounded in any empirical data, and in tendency fails to acknowledge that also along the diagonal, there was no persistent effect beyond 500 ms pre-word onset.With respect to the neural mechanism, the authors state that "the absence of an effect off the diagonal suggests that the spatial patterns associated with pre-activation evolved dynamically over time". However, there is no evidence to support this claim. In particularly when considering that there is also no persistent effect along the diagonal, it most likely indicates that there was no sustained pre-activation over time.

As noted above, we have acknowledged that the interpretation of the dynamically evolving spatial patterns remains speculative.

Reviewer #2:This manuscript presents research aimed at investigating the hypothesis that specific words are pre-activated in the brain given a constraining semantic context. The authors test this hypothesis by presenting highly constraining sentences such that the final word in each sentence can be easily predicted. Moreover, they do so such that pairs of sentences are likely to be predicted to finish with the same word. They then examine the similarity of spatial and temporal patterns in MEG preceding the presentation of the final words. In particular they compare the similarity in these patterns between pairs of sentences with the same final predicted word and pairs of sentences with different final words. They find that both the spatial patterns and temporal patterns are MORE similar for sentences where the same final word is predicted than for sentences where different final words are predicted. They take this as evidence that specific lexico-semantic predictions are made by the brain during language comprehension.This was a very well designed piece of research with interesting and compelling results. The manuscript was well written and the discussion seemed reasonable.I have a few relatively minor comments and queries:1) The nice study design included ensuring that the paired sentences didn't actually finish in the same word and that sometimes the sentence with an unexpected word would appear first and sometimes the sentence with the expected word would appear first. The authors argue that this means the results are not simply explainable on the basis that subjects might retain the expected final word in memory when reading the second sentence of a pair. However, it seems to me that, even though the unpredicted word has a much lower cloze, the subject might still retain that unexpected word in memory when hearing the second sentence of a pair. It doesn't seem that likely to me, but it's conceivable. I mean when a subject hears the unexpected word 'child', they might be more likely to retrieve that word when they are next presented with a sentence for which 'baby' is the "correct" prediction, but for which 'child' is a reasonable final word. So, much and all as I like the design, I do think it is still possible that retrieval of a previously stored word is still possible. One thing that I was unclear on (and sorry if I just missed it) was the actual ordering of the sentence presentation. Did the two members of a pair of sentences always appear consecutively? If so, this would make the idea of retrieval even more likely. If the 120 sentences are all just presented in a random order, then I guess it is unlikely. Again, sorry if I missed that.

A) We apologize for not making this clearer in the previous version of the manuscript. In the revised Introduction, we now clarify that “During the experiment, sentences were presented in a pseudorandom order, with at least 30 other sentences (on average 88 sentences) in between each member of a given pair.” In the Results, we state that “The sentences were constructed in pairs (120 pairs) that strongly predicted the same sentence-final word (SFW), although, during presentation, members of the same pair were separated by at least 30 (on average 88) other sentences.” In the Material and methods, we state that “the two members of each pair were presented apart from each other, with at least 30 (on average 88) sentences that predicted different words in between.”

B) We now discuss the possible influence of the order of the presentation of the two members of each sentence pair in the revised Discussion: “A second set of alternative interpretations might acknowledge that the increase in spatial similarity detected in the *within-pair* sentencesreflects activity related to the prediction of a specific SFW. However, instead of attributing the effect to the predicted representation itself, they might attribute it to participants’ *recognition* of a match between the word that they had just predicted and a word that they had actually seen as the SFW earlier in the experiment. This seems unlikely because we found that the spatial similarity effect was just as large when the unexpected SFW of a pair was presented before the expected SFW, as when the expected SFW was presented first (see Figure 2—figure supplement 3). It is, however, conceivable that participants recognized a match between the word that they had just predicted and a word that they had predicted earlier in the experiment (even though this predicted word was never observed). For example, there is some evidence that a predicted SFW can linger in memory across four subsequent sentences, even if it is not actually presented (Rommers and Federmeier, 2018). This seems less likely to have occurred in the present study, however, where each member of a sentence pair was separated by at least 30 (on average 88) other sentences.”

2) A minor query – were there different numbers of words in the sentences? Or always the same? And, relatedly, did the subject always know when the final word was going to appear? It's just that a pet worry of mine is the generalizability of language research done on isolated sentences that are very regular in their makeup. I imagine subjects get into an unusual mindset with linguistic processes overlapping with more general decision making strategies that may confound things. I don't think that's an issue here for two reasons: 1) it wouldn't explain why the data are more similar within sentences than between and 2) subject didn't have to make deliberative decisions at the end of each sentence. But still, it would be nice to get a sense of the variability (or lack of it) in the structure of the sentences.

In the Materials and methods, we have provided more information on the length of the contexts up until and including the SFW-1 (ranging from 4 to 12 words). Thus, to address the reviewer’s question, the lengths of these sentences were quite variable, ranging from 5 to 13 words. Therefore, participants wouldn’t have known when the SFW was going to appear.

Also, we now make it clear that the difference in the number of words was matched between pairs that constrained for same word (*within-pairs*) and pairs constrained for a different word (*between-pairs*): t_(28678)_ = -1.26, p = 0.20.

3) Very minor – in subsection “Design and development of stimuli” there seem to be 109 pairs of sentences above 70% close and 12 that were lower. That makes 121, not 120.

We thank the reviewer for pointing this out. We have clarified that *11 pairs* of sentences had cloze values that were lower than 70%.

4) In subsection “MEG data processing” the authors say "Within this 4000ms epoch, trials contaminated…were…removed…" How is there a trial within this epoch? Is the trial not the entire epoch? Or am I misunderstanding what you mean by a trial?

We apologize for the confusion. The trial refers to the entire epoch. In the Materials and methods, we have changed the sentence to “Trials (i.e. whole epochs) contaminated with muscle or MEG jump artifacts were identified and removed using a semi-automatic routine.”

Reviewer #3:My two main concerns are as follows. First, that the specific method used for the RSA analysis -separating spatial and temporal dimensions of the data and using one dimension (spatial) to narrow down the testing time-window of the other (temporal) also narrowed down the scope of the effects uncovered. Second, that the sentences within- and between-pairs were insufficiently matched in terms of the syntactic and lexicosemantic characteristics of the words directly preceding the critical predicted word and observed effects could have been related to those differences rather than pre-activations. These two issues would need to be addressed before the strength and interpretation of the current set of effects could be fully evaluated.Major points:1) My main concert in terms of analysis methods is that the separation of the temporal and spatial components of the RSA analysis unnecessarily limited the kind of effects that were uncovered. For calculation of both spatial similarity time-series and cross-temporal spatial similarity matrixes all MEG sensors were included and hence the effects would be greatest in the time-points where many sensors simultaneously show similar activity for pairs of sentences. This means that strong and extended in time but spatially localised (or insufficiently distributed) effects might be missed. Especially since for determining the significant time-windows, vectors were averaged across subjects, which means that localised effects had even less chance of surviving given that the same effects in different subjects could appear in different sensors – due to the differences in head shape etc. Further the concern for the temporal RSA is that the time-window where the effects were tested (-880 -485, SFW aligned) were derived on the basis of the spatial analysis, where spatially continuous differences between within- and between-pair similarity values were found after applying an arbitrary cut-off (r>0.04).To avoid these issues a spatiotemporal RSA could be carried out in the source space directly, or firstly in the sensor space (across sensors and time points) and then the significant spatiotemporal clusters could be source localised. For example, if beamforming is used to derive single-trial source estimates, then data RDMs can be derived using a modified version of the Searchlight approach (e.g Nili et al., 2014; Su et al., 2012 and 2014). For every trial, at every grid point and every time step (every 1/5/10 ms) a 3D data matrix is extracted consisting of activation from n of neighbouring grid points and n time-samples. Then for each pair of sentences predicting the same trials these data matrixes are correlated. Then the effects are averaged across all within-pairs producing grid point by time point spatiotemporal correlation values for the within-pair condition. The same can be repeated for between-pairs. Then a pair t-test can be done to compare within- and between- data across time and grid space, significant spatiotemporal clusters of differences would be determined with cluster permutation. If no major effects have been missed by separating spatial and temporal dimensions of the data, then spatiotemporal RSA would further validate the current set of results.

We thank the reviewer for the suggestion.

A) In the Results, we have pointed out that the analysis approach that we took is fairly conservative: “it was limited to the time window that showed a spatial similarity effect, and so it may not have captured more extended temporal similarity effects that were not accompanied by a spatial similarity effect”. As pointed out by the editor, “this methodological concern could explain the absence of some effects in the existing analyses, but not the presence of reliable effects”. Also, “substantial correction for multiple comparisons in the searchlight analysis might reduce sensitivity”. Therefore, we decided not to take this approach in the current study. Rather, “we were interested, *a priori,* in any functional relationship between these measures, i.e. whether the spatial similarity effect reflected brain activity associated with the prediction of spatially distributed semantic representations, and whether the temporal similarity effect reflected brain activity associated with temporal binding of these spatially distributed representations.”

However, we state that “in order to fully exploit the spatiotemporal pattern of the data, future studies could examine the spatial and temporal patterns simultaneously using a spatiotemporal searchlight approach (Nili et al., 2014; Su et al., 2012; Su et al., 2014).”

2) I have several questions about the experimental stimuli. Firstly, were the experimental sentences both between and within-pairs controlled for sentence length (n of words) and syntactic complexity (n of clauses, presence of embedded dependences)? The issue would arise if, for example, all within-pairs happened to have the same syntactic structure/complexity, while between-pairs had mismatching or different structure/complexity. Then the increases of the similarity before SFW for the within-pairs could potentially be attributed to similar demands of grammatical/syntactic processing, while decreased similarity for between-pairs would be driven by differences in these processing demands. The authors cover this potential caveat in subsection “Unique spatial patterns of neural activity are associated with the prediction of specific words, prior to the appearance of new bottom-up input” of the discussion, and argued that in this case we would see within- and between-pair difference arise earlier. However, while such differences could have been building up, they also could have become significant only closer to the end of the sentence. To exclude this option differences between within- and between-pair sentences should be reported.Secondly, were the SFW-1 words (the word directly before the SFW) controlled for any of the following characteristics across conditions: syntactic class, frequency, any semantic characteristics such as imageability, concreteness? Again, if the within-pairs matched in terms of SFW-1 characteristics more than the between-pairs sentences effects in the 'prediction' time-window could be driven by similarities of the SFW-1 processing and not the by the SFW pre-activation. Since the critical claim of this paper is that increases in spatial and temporal correlation of the neuronal activity for the averaged within-pairs is driven by pre-activation of the SFW it is critical to exclude any of the effects described above.

We fully agree that it is very important to rule out other factors that could lead to a greater similarity in brain activity on the SFW-1 in the *within-pair* sentences than the *between-pair* sentences. We have made the following major changes to the manuscript to address this:

A) In the Materials and methods, we now state that we measured: (1) the number of words, the number of clauses, and the syntactic complexity of the sentence contexts up until and including SFW-1; (2) various lexical properties of the SFW-1 itself (i.e. visual complexity, word frequency, syntactic class); and (3) the predictability (as operationalized by cloze probability) of the SFW-1. We showed that none of these factors differed systematically between pairs of contexts that predicted the same SFW (i.e. *within-pairs*) and pairs of contexts that predicted a different SFW (i.e. *between-pairs*).

In Chinese, it is difficult to measure the orthographic or phonological features of the SFW-1 as a whole. This is because the characters within each word/phrase of the SFW-1 had distinct orthographic and phonological features. Also, as shown in the full set of stimuli (Figure 1A—source data 1), the SFW-1 could either be a content word (verb, noun, adjective, adverb) or a function word (pronoun, classifier, conjunction, particle, prepositional phrases). This makes it difficult to examine the concreteness or imageability of the SFW-1 in all sentences (there is no available Chinese corpus listing all these words). However, given the heterogeneity of the SFW-1, we think that these factors are unlikely to have influenced the observed effect.

B) We carried out a new control analysis that aimed to fully exclude the possibility that the spatial similarity effect was driven by bottom-up processing of the SFW-1 rather than by anticipatory processing of the SFW itself. In this control analysis, we selected a subset of *between-pair* sentences that contained exactly the same SFW-1, but nonetheless predicted a different SFW. Then we selected sentences that constrained for these same SFWs (*within-pairs)*, but which differed in the SFW-1. We then compared the spatial similarity between these two subsets of sentence pairs. If the increased spatial similarity associated with the *within-pairs* versus *between-pairs* was due to lexical processing of the SFW-1, then the spatial similarity should be greater in sentence pairs containing exactly the same SFW-1 (i.e. in the subset of *between-pairs*) than in sentence pairs that predicted the same SFW (i.e. in the subset of *within-pairs*). We found no evidence for this. Instead, the spatial similarity remained larger for the *within-pairs* than the *between-pairs* (although in this subset analysis, the difference only approached significance due to the limited statistical power).

C) In the Discussion, we now explicitly discuss why the spatial similarity effect cannot be explained by the contexts of the sentence pairs or the lexical properties of the SFW-1.

3) This point is related to the conclusions drawn by the authors in the Discussion section about the nature of the pre-activated representations. The authors suggest that the effects observed in the pre-SFW window can be driven by orthographic or phonological features of the predicted words. Have any of the analyses they proposed (subsection “Unique spatial patterns of neural activity are associated with the prediction of specific words, prior to the appearance of new bottom-up input”) been carried out? Since sentences used for this study were indeed very constraining, SFW pre-activation of the perceptual features of strongly predicted words would be expected under the predictive processing/coding approach.

In the Discussion, we have explained why the current study cannot address the question of whether or not the observed effects reflect the prediction of just the semantic features or also the orthographic or phonological features of the predicted words: “It is also possible that the increased spatial similarity in association with sentence pairs that predicted the same word reflected similarities of predictions generated at a lower phonological and/or orthographic level of representation. On this account, comprehenders not only predicted the semantic features of words, but they also pre-activated their word-forms. The present study cannot directly speak to this hypothesis. This is because, for the most part, there is a one-to-one correspondence between the semantic features and the phonological or orthographic forms of words. However, the methods described here provide one way of addressing this question in future studies. For example, by examining the spatial similarity of sentence pairs that constrain for words that share orthographic features but that differ in their meanings (homonyms), it should be possible to dissociate the prediction of orthographic/phonological representations from the prediction of semantic features associated with a given lexico-semantic item.”

[Editors' note: further revisions were requested prior to acceptance, as described below.]

While the manuscript has been much improved there are two remaining issues that I think need to be addressed before acceptance, as outlined below:1) There are too many places in the Introduction and Discussion in which I think the authors aren't thinking critically enough about whether it is only their preferred "generative and predictive" view that could explain the present findings. My view is that many other accounts could also explain their findings. Specifically, any model which: (i) activates a cumulative semantic representation of sentence meaning, and (ii) emphasises processing speed and efficiency such that semantic representations that are strongly implied by the words read so far, but not yet directly expressed in words are activated – can also account for the current findings. There are many such models in the literature, but most notable (to my mind) is the "sentence gestalt" model from St John and McClelland, 1990 that has been recently updated by Rabovsky et al., 2018, and can predict the magnitude of EEG N400 responses in a wide range of sentence processing paradigms. To my knowledge this is not a model which is explicitly "generative and predictive" and yet I think it very likely that RSA analysis of the sentence gestalt representations generated by this model could simulate the results of the present study. While I don't think that the authors need to do the work to explore whether the model *can* simulate their findings, I do think that it is in their interests to offer a more balanced overview of the literature and to more precisely explain what sort of computational model is implied by their findings.

Thank you for bringing up these points.

We agree that the idea that the brain predicts semantic features associated with specific words does not follow specifically from the type of generative framework of language comprehension sketched out in section 5 of Kuperberg and Jaeger, 2016 or by Kuperberg, 2016. In the revised Introduction, we have removed all mention of a generative framework. Rather, we simply state, “Prediction is hypothesized to be a core computational principle of brain function (Clark, 2013; Mumford, 1992). During language processing, probabilistic prediction at multiple levels of representation allows us to rapidly understand what we read or hear by giving processing a head start (see Kuperberg and Jaeger, 2016, for a review).” (Note that we cite Kuperberg and Jaeger here as a comprehensive review of a large literature on prediction at multiple levels of representation — we only discussed the generative framework in the final section of that paper).

In the Discussion, in response to a reviewer, we brought up the generative framework more specifically to explain the *earliness* of the spatial and temporal similarity effects: the prediction of the SFW was generated at the first point in time at which participants had sufficient information to unambiguously generate this prediction, which was after the onset of the penultimate word. We suggested that, in the sentence “In the crib, there is a sleeping …”, as comprehenders accessed the meaning of the word, <sleeping>, they may have also predicted the semantic features of <baby>.

We argued that “this type of account follows from a generative framework of language comprehension in which, following highly constraining contexts, comprehenders are able to predict entire events or states, along with their associated semantic features…”. Here, we referenced Kuperberg and Jaeger, 2016 (referring to section 5) as well as Kuperberg, 2016 — papers in which we had outlined what this type of framework might look like at Marr’s *computational level* of analysis. The recent paper by Rabovsky, Hansen and McClelland, 2018, and its predecessor (St John and McClelland, 1990) describe models implemented at Marr’s *algorithmic level* of analysis. They share similar assumptions to those outlined by Kuperberg and Jaeger, 2016 section 5 and Kuperberg, 2016. We now include both citations at this point.

Regarding the editor’s note that the latter two models “are not explicitly generative and predictive": As discussed by McClelland, 2013, many connectionist models, although implemented at Marr’s algorithmic levels of analysis, are inherently generative, and probabilistically predictive, with close links to probabilistic Bayesian frameworks. The model by Rabovsky, Hansen and McClelland, 2018, probabilistically infers hidden causes (events) after encountering sequential inputs, and it is therefore both generative and probabilistically predictive. Indeed, it is characterized as such at the beginning of the Materials and methods: “The model environment consists of [sentence, event] pairs probabilistically generated online during training according to constraints embodied in a simple generative model”. The framework outlined in section 5 of Kuperberg and Jaeger is similarly generative and probabilistically predictive, and we believe that the two frameworks share many core assumptions.

There is, however, perhaps one relevant difference in the assumptions of the probabilistic framework outlined by Kuperberg, 2016, and the model implemented by Rabovsky, Hansen and McClelland, 2018: Kuperberg, 2016, is clear that the N400 primarily reflects the (subjective) probability of semantic features associated with an input (word or other stimulus), given the probability distribution over the latent cause (events) inferred just before the semantic features of the incoming word become available from the bottom-up input. Rabovsky, Hansen and McClelland, 2018, however, do not explicitly include a semantic features layer in their model. On the other hand, they do include statements that suggest that what they are indexing is, in fact, changes in activity at the level of semantic features associated with an input word, given the event predicted by the preceding context (e.g. “The N400 corresponds to the amount of unexpected semantic information in the sense of Bayesian surprise”; the model “provides a basis (together with connection weights in the query network) for estimating these probabilities [of semantic features] when probed.”). It is currently unclear to us whether this simply amounts to a difference in modeling approach, or whether this amounts to a true difference in assumptions about architecture.

As regards the current set of findings, however, we find it helpful to understand the effects observed as reflecting commonalities in the predicted *semantic features* associated with the prediction of specific words, rather than *purely* reflecting similarities at the level of the entire event (or shift in state to get to this event). In other words, while it may be that the representation of semantic features associated with an individual word are inherently tied in with the event being inferred, we still find it helpful to refer to “semantic features” associated with this word descriptively, both in relation to the N400 as well as in relation to the current findings. To be more specific, in the paired sentences, “In the crib there is a sleeping…” and “In the hospital, there is a newborn…”, we think that the increased *within-pair* spatial similarity effects observed ultimately reflected the predicted semantic features associated with <baby>, rather than the similarities between the two predicted events as a whole: the <baby sleeping in the crib> event and the <newborn baby in the hospital> event. These two events are distinct *except for* the presence of the semantic features, <baby>. We have therefore added an additional sentence to make this clear, and here we reference Kuperberg, 2016, who is explicit in discussing the N400 as reflecting the probability of encountering a given set of semantic features, given the agent’s current probabilistic beliefs about event being communicated.

This section in the Discussion now reads as follows:

“…. This provides evidence that the prediction of the SFW was generated at the first point in time at which participants had sufficient information to unambiguously generate this prediction. For example, in the sentence “In the crib, there is a sleeping …”, as comprehenders accessed the meaning of the word, <sleeping>, they may have also predicted the semantic features of <baby>. This type of account follows from a generative framework of language comprehension in which, following highly constraining contexts, comprehenders are able to predict entire events or states, along with their associated semantic features, prior to the appearance of new bottom-up input (e.g. Kuperberg and Jaeger, 2016, sections 4 and 5; Kuperberg, 2016; St John and McClelland, 1990; Rabovsky, Hansen and McClelland, 2018). Importantly, however, we conceive of the *within-pair* spatial similarity effect detected here as primarily reflecting similarities at the level of semantic features (e.g. <human>, <small>, <crying>) associated with the predicted word (“baby”), rather than similarities between the entire predicted events (e.g. the <baby sleeping in the crib> event versus the <newborn baby in the hospital> event) (see Kuperberg, 2016). As noted above, we cannot tell from the current findings whether this, in turn, led to the top-down pre-activation of specific phonological or orthographic word-forms.”

Earlier in the Discussion, we made it clear that “It is also possible that the increased spatial similarity in association with sentence pairs that predicted the same word reflected similarities of predictions generated at a lower phonological and/or orthographic level of representation. On this account beliefs about the underlying event and semantic features led to the top-down pre-activation of information at these lower levels of the linguistic hierarchy before new bottom-up information becomes available (see Kuperberg and Jaeger, 2016, sections 3 and 5 for discussion). The present study cannot directly speak to this hypothesis.” Note that here we referred only to Kuperberg and Jaeger, 2016, sections 3 and 5, which, unlike Rabovsky, Hansen and McClelland, 2018, *does* assume a hierarchy and clear representational distinctions between events and phonological/orthographic word form.

2) I had one other minor question about the method that they used in comparing cloze probabilities between and within item pairs which could be addressed by same time. This point is described in more detail in their rebuttal letter than in the manuscript. However, I think that this issue deserves a little more attention in the manuscript given the known importance of cloze probability in predicting the magnitude of EEG/MEG signals during sentence processing, and the. Specifically, in the rebuttal letter the authors report analyses of the difference between cloze probability for sentence pairs. However, if my understanding of this analysis is correct this analysis should be conducted not on the difference between cloze probabilities, but rather the absolute difference between cloze probabilities for within and between item pairs. I think that otherwise the average difference between cloze values would always be zero. I'd like the authors to report this analysis in the manuscript, including a description of the method used for conducting the analysis.

We apologize for the confusion. In the Materials and method session of the manuscript, we now clearly describe how the analysis was conducted: “for each possible pair of sentences, we calculated the absolute difference in the cloze probability of the SFW-1 and carried out an independent sample t-test. Any differences in cloze probability were matched between pairs that constrained for the same word (*within-pairs:* 17.00% cloze difference) and pairs that constrained for a different word (*between-pairs*: 17.28% cloze difference), t_(28678)_ = -0.136, p = 0.89.”